# Pedogenic and microbial interrelation in initial soils under semiarid climate on James Ross Island, Antarctic Peninsula Region

Lars A. Meier[1*], Patryk Krauze[2*], Isabel Prater[3], Fabian Horn[2], Carlos E.G.R. Schaefer[4], Thomas Scholten[1], Dirk Wagner[2, 5], Carsten W. Mueller[3,6], and Peter Kühn[1]

[1]Department of Geosciences, University of Tuebingen, Tuebingen, D-72070, Germany
[2]GFZ German Research Centre for Geosciences, Section Geomicrobiology, Potsdam, D-14473, Germany
[3]Lehrstuhl für Bodenkunde, TU München, Freising, D-85354, Germany
[4]Departamento de Solos, Universidade Federal de Viçosa, Viçosa, BR-36571-000, Brazil
[5]Institute for Earth and Environmental Sciences, University of Potsdam, Potsdam, D-14476, Germany
[6]School of Agriculture and Food Sciences, The University of Queensland, St Lucia, Queensland, AU-4072, Australia

*Correspondence to*: Lars A. Meier (lars-arne.meier@uni-tuebingen.de)

*shared first authorship

**Abstract.** James Ross Island (JRI) offers the exceptional opportunity to study microbial driven pedogenesis without the influence of vascular plants or faunal activities (e.g. penguin rookeries). In this study, two soil profiles from JRI (one at St. Martha Cove - SMC, and another at Brandy Bay - BB) were investigated, in order to gain information about the initial state of soil formation and its interplay with prokaryotic activity, by combining pedological, geochemical and microbiological methods. The soil profiles are similar in respect to topographic position and parent material but are spatially separated by an orographic barrier and therefore represent windward and leeward locations towards the mainly south-westerly winds. These different positions result in differences in electric conductivity of the soils caused by additional input of bases by sea spray at the windward site, and opposing trends in the depth functions of soil pH and electric conductivity. Both soils are classified as Cryosols, dominated by bacterial taxa such as Actinobacteria, Proteobacteria, Acidobacteria, Gemmatimonadates and Chloroflexi. A shift in the dominant taxa was observed below 20 cm in both soils as well as an increased abundance of multiple operational taxonomic units (OTUs) related to potential chemolithoautotrophic Acidoferrobacteraceae. This shift is coupled by a change in microstructure. While single/pellicular grain microstructure (SMC) and platy microstructure (BB) is dominant above 20 cm, lenticular microstructure is dominant below 20 cm at both soils. The change in microstructure is caused by frequent freeze-thaw cycles and a relative high water content and goes along with a development of the pore spacing and is accompanied by a change in nutrient content. Multivariate statistics revealed the influence of soil parameters such as chloride, sulfate, calcium and organic carbon contents, grain size distribution, and pedogenic oxide ratios on the overall microbial community structure and explained 49.9% of its variation. The correlation of the POR with the compositional distribution of microorganisms as well as the relative abundance certain microorganisms such as potentially chemolithotrophic Acidiferrobacteraceae-related OTUs could hint on an interplay between soil forming processes and microorganisms.

## 1 Introduction

In extreme environments, like Antarctica, local climatic conditions such as low temperatures, precipitation or irradiance are important and often limiting factors for soil formation. Even though soils in Antarctica are often poorly developed, they can be highly diverse (Michel et al., 2014; Simas et al., 2008; Bockheim et al., 2015). Therefore, soil scientific investigations became a relevant topic in Antarctic research, proving that there are actually soils in Antarctica (Jensen, 1916) and identifying soil forming processes (Ugolini, 1964). Antarctic soil research is mostly located in Victoria Land, continental Antarctica, especially in the McMurdo Dry Valleys (Michel et al., 2014; Ugolini and Bockheim, 2008), in the South Shetlands, maritime Antarctica (Simas et al., 2015) and the western Antarctic Peninsula Region (APR) (Haus et al., 2015; Hrbáček et al., 2017b; Schaefer et al., 2017; Souza et al., 2014; Pereira et al., 2017).

Soils on continental Antarctica are often saline with thick salt horizons (Souza et al., 2014). Due to environmental stressors such as very low temperatures, low water availability, frequent freeze-thaw cycles and limited organic nutrient contents, soils from continental Antarctica show limiting conditions for higher organisms (Cary et al., 2010). However, diverse microbial communities thrive in a variety of Antarctic habitats, such as permafrost soils (Cowan et al., 2014).

Soils in maritime Antarctica and western APR differ from soils in continental Antarctica according to their stage of development (Balks et al., 2013; Blume et al., 2004; Parnikoza et al., 2017). They show extensive cryoturbation processes with occasional salt crusts at the soil surface (Balks et al., 2013; Bockheim, 1997). Local conditions determine nutrient availability in Antarctic soils (Prietzel et al., 2019). Ca, Mg, K and P contents are generally high in igneous and volcanic rocks, whereas P and N contents are highest in ornithogenic soils. Ornithogenic soils are well known in Antarctica. The World Reference Base for Soil Resources (WRB, 2014) defines ornithogenic material (from Greek ornithos, bird, and genesis, origin) as material, which is characterized by penguin deposits mainly consisting of guano and often containing a high content of gravel transported by birds (cf. Ugolini, 1972). Soils from the eastern part of the APR (also called Weddell Sea sector) are different, since they are associated with a dry climatic transitional zone between the wet, warmer maritime Antarctica and colder, arid continental Antarctica. Mean temperatures are below 0°C and liquid water supply is sufficient to allow soil forming processes (Souza et al., 2014). Souza et al. (2014) also showed that cryoturbation is less pronounced in the eastern APR than in the South Shetlands. The base saturation (>50%) and electric conductivity (EC) are generally high whereas the amount of total organic carbon (TOC) is substantially low. Regarding cryoturbation, active layer depth, chemical weathering

and soil organic C-content, soils from the eastern APR are comparable to soils from inland areas of the Ross Sea Region (Balks et al., 2013), though they are formed on different parent material (Daher et al., 2018). In comparison, eastern APR with its semiarid soils remains one of the least studied areas in Antarctica (Souza et al., 2014; Daher et al., 2018).

Since Microorganisms in Antarctica show a broad diversity as revealed by recent molecular phylogenetic and metagenomic methods (Cowan et al., 2014) and contribute to the weathering of minerals in soils (Uroz et al., 2009), they are pivotal to understand initial soil formation. The bacterial phyla Proteobacteria, Acidobacteria, Actinobacteria, Bacteroidetes, Firmicutes and Gemmatimonadates, commonly found in temperate soils, also dominate the microbial communities observed in Antarctic habitats (e.g. Bajerski and Wagner, 2013; Cary et al., 2010; Pearce et al., 2012; Chong et al., 2012). The microbial community structure is influenced by local soil chemical parameters, especially pH (e.g. Chong et al., 2010, Siciliano et al., 2014), but also by soil physical parameters such as grain size distribution and soil moisture (Ganzert et al., 2011). Chong et al. (2015) proposed, however, that historical contingency and dispersal limitations could have a stronger influence on differences in community distributions at a regional scale (>1000km). At the microscale, microbial activity such as photosynthesis and nitrogen fixation has a distinct influence on soil chemical parameters, e.g. the increase of carbon and nitrogen contents in oligotrophic soils (Ganzert et al., 2011; Cowan et al., 2011; Niederberger et al., 2015). In return, these changes in soil characteristics affect microbial community composition. Conflicting results illustrate the lack in the understanding of drivers of soil microbial diversity in high latitude soils (Cowan et al., 2014). Since most of the non-lichenized Antarctic fungi are known to be decomposers and their abundance and distribution is limited by plant derived nutrients, and bio-available Carbon (Arenz et al., 2011), the focus of this study lies on the prokaryotic interplay with soil characteristics and soil formation.

Micromorphological studies in the maritime Antarctica and the western APR described sulphurization and phosphatization in ornithogenic soils and mineral transformation on volcanic rocks (Pereira et al., 2013; Schaefer et al., 2008); and paleosols (Kirshner and Anderson, 2011; Spinola et al., 2017). Even though micromorphology offers the opportunity to study constituents of soil and their mutual relations in space and time and to identify soil forming processes in an undisturbed state (Stoops, 2003), so far no micromorphological study has been published about soil forming processes in the eastern APR that are influenced neither by sulfates nor by birds.

Our study sites are located on James Ross Island in the eastern APR (Fig.1) and therefore offer a unique setting to study soil formation and microbial communities in a transitional Antarctic

landscape between the wet maritime and dry, colder continental Antarctica. We selected two different soils, representing coastal soils and inland soils of James Ross Island, developed on similar substrate and at similar topographic positions, but differing in local climate conditions and nutrient contents due to their relative position towards the mainly SW-winds. The western study site (Brandy Bay –BB) is located in a windward position and is highly influenced by sea spray, while the eastern study site (Santa Martha Cove – SMC), located behind a mountain range, is located in a leeward position (Prietzel et al., 2019). This setting enables an investigation of interdependencies particularly between prokaryotic life and soil properties, since the selected soils are not influenced by vascular plants, sulfides, and penguin rookeries. With this, the main goal of our study is to identify major soil and microbiological properties in an extreme environment by combining pedochemical and micromorphological methods with microbial community studies based on high throughput sequence analyses. Thus, we will gain a better general understanding of (i) the main soil forming processes and (ii) the drivers of soil microbial diversity community structure in the eastern APR. This addresses also the question, if the variance of pedogenic and microbiological properties are larger between depth increments within one profile (e.g. with different distances to the permafrost table) or between different soil profiles, i.e. due to different local environmental conditions.

## 2. Material and Methods

### 2.1. Regional setting of James Ross Island, maritime Antarctica

**[ Figure 1 ]**

James Ross Island is situated east of the Antarctic Peninsula and is the largest island in the western Weddell Sea sector (Hjort et al., 1997). The study area is located on Ulu Peninsula in the northern part of JRI (Fig. 1). It represents one of the largest ice-free areas of the APR (Nedbalová et al., 2013; Hrbáček et al., 2017b) with the beginning of its deglaciation 12.9 ±1.2 ka ago (Nývlt et al., 2014). More than 300 km² of the JRI lowlands are currently ice-free, except for a few glaciers (Engel et al., 2012).

The climate on JRI is semi-arid polar-continental (Martin and Peel, 1978). The precipitation, mostly snow, ranges between 200 to 500 mm of water equivalent per year with the major share during winter (Davies et al., 2013; Zvěřina et al., 2014). The thickness of the snow cover does not exceed 30 cm, but varies due to strong winds (Hrbáček et al., 2017b; Hrbáček et al., 2016a). The annual air temperature ranges between +10 °C and -30 °C on Ulu Peninsula (Hrbáček et al., 2016a; Láska et al., 2011). The year 2015 marked the warmest summer ever measured on

Ulu Peninsula, having a mean seasonal summer temperature (MSST) of 0.0 °C and a maximum air temperature of 13.3 °C (Hrbáček et al., 2017a); even though the mean annual air temperature (MAAT) decreased slightly from -6.8 °C in 2011 to -7°C in 2015 (Hrbáček et al., 2016b; Láska et al., 2012).

The two study sites are located at Brandy Bay (BB) near the western coast and at St. Martha Cove (SMC) at the eastern coast of Ulu Peninsula. Both sites are located at similar topographic positions (small plateaus) and elevation (80 m a.s.l.) with no visible vegetation (Fig. 2 and Fig. 3).

**[ Figure 2 ]**

BB is located windward towards the mainly south-westerly winds (Hrbáček et al., 2016c; Nývlt et al., 2016), whereas SMC is located leeward, shielded by the Lachman Crags from the stronger winds. This results in less precipitation in the eastern part of JRI (Davies et al., 2013). Therefore, BB can be considered as a characteristic wind-exposed coastal site with high influence of sea spray, whereas SMC represents a characteristic soil of an inland site with less influence of sea spray.

The substrate of both study sites is basically composed of coarse-grained cretaceous sandstones and siltstones of the Alpha Member of the Santa Martha Formation (Hrbáček et al., 2017b). The land surface is generally covered by a debris layer of gravels and large clasts mixed with loose sandy regolith, mostly derived from James Ross Volcanic Group basalts, which were deposited as debris flows containing mainly basalt and hyaloclastite breccia and palagonite (Davies et al., 2013; Hrbáček et al., 2017b; Salzmann et al., 2011). No nesting birds are found on JRI.

The continuous permafrost on James Ross Island shows an active layer thickness ranging between 40 and 107 cm related to the topographic position on Ulu Peninsula (Bockheim et al., 2013; Borzotta and Trombotto, 2004).

**2.2 Soil sampling**

During the austral summer period in 2016 soil samples from BB and SMC (Fig. 4 and Fig. 5) were taken. Both profiles were dug until a layer of coarse gravel was found. Bulk samples of both profiles were taken in depth increments (0-5cm, 5-10cm, 10-20cm, 20-50cm, >50cm) and were placed into sterile plastic bags, which were frozen immediately. Continuous cooling at -20°C was ensured by a transfer with the research vessels *RV Polarstern* to Germany. For micromorphological analyses, undisturbed and oriented samples were taken in modified

Kubiena boxes (10cm x 6cm x 5cm). Samples for micromorphology were taken at depth of 0-
10cm, 10-20cm, 30-40cm, 50-60cm and 80-90cm at SMC. BB samples represent the depth of
10-20cm, 20-30cm und 40-50cm. Soils were described according to Food and Agriculture
Organization of the United Nations (FAO) (2006) and classified according to the World
Reference Base for Soil Resources (WRB; IUSS Working Group WRB, 2015).

**2.3 Soil physical and chemical analysis**

**2.3.1 Grain size distribution**

The samples were saturated (100ml of deionized water) and sonicated (800J ml$^{-1}$). Coarse-
medium sand (>200µm), fine sand (63-200µm) and coarse silt (20-63µm) were obtained by wet
sieving. The smaller fractions, including medium silt (6.3-20µm), fine silt (2-6.3µm) and clay
(<2µm), were separated by sedimentation. Fractions >20µm were dried at 45°C and weighed
afterwards. The fractions <20µm were freeze-dried before weighing. The different procedures
were chosen due to practical reasons: freeze-drying allows submitting the finer fractions to
further analyses (particularly carbon and nitrogen content) immediately, while the coarser
fractions need milling anyway.

**2.3.2 pH, EC, C&N contents, major elements and pedogenic oxides**

The pH value was obtained using a pH meter (ph197i, WTW, Germany). Electrical conductivity
was measured with a conductivity meter (LE703, Mettler-Toledo, USA). Values of pH and
electric conductivity were measured from bulk samples < 2mm in deionized water with a
sample to water ratio of 1:2.5.
Carbon (C) and nitrogen (N) contents of the bulk soils were analyzed by dry combustion
(Elementar CNS Vario Max Cube). 300 to 500mg per sample were analyzed in duplicate. In
Order to distinguish between the total organic carbon (TOC) content and the total inorganic
carbon (TIC), TIC was removed by acid fumigation after Ramnarine et al. (2011). 100 mg of
the milled bulk soil samples were moistened with 20 to 40 µl of deionized water and put into a
desiccator together with 100ml of 37% HCl. Afterwards, the samples were dried at 40°C.
Finally, the samples were measured again by dry combustion (EuroVector EuroEA3000
Elemental Analyser) to obtain the TOC content. TIC content was calculated: TIC $= C_{tot}$ - TOC
Major elements were analysed with a wavelength dispersive XRF device (AXS S4 Pioneer,
Bruker, USA). Prior to preparation, the samples (ratio Li-metaborate to soil 1:5) were ground
with an agate mill for 12 minutes. Major elements were used for the calculation of weathering
indices.
Pedogenic iron-oxides ($Fe_d$) were determined by dithionite-citrate-hydrogen carbonate
extraction (Holmgren, 1967). Poorly to non crystallised Fe-oxides ($Fe_o$) were determined by
acid ammonium extraction (Schwertmann (1964). The extractions were analysed at a
wavelength of 238.204 nm by an inductively coupled plasma optical emission spectrometer
(Vista Pro CCD Simultaneous ICP-OES, Varian, USA).

### 250 2.3.3 Ion chromatography

The initial water content in the investigated soil material was too low to extract sufficient
amounts of pore water for ion chromatography. Hence, the soil samples were leached, according
to Blume et al. (2011). Five grams of soil material were suspended in 25ml deionized water,
shaken for 90 minutes and centrifuged at 9000rpm to separate the soil material from the soil
solution and sterile filtered through a 0.22µm PES filter (Sartorius AG, Germany).
The ion concentrations in leached water samples were analysed by using two ion
chromatography (IC) systems (SYKAM Chromatographie Vertriebs GmbH, Germany). For
cations, the IC system consisted of a 4.6 x 200 mm Reprosil CAT column (Dr. Maisch HPLC
GmbH, Germany), an S5300 sample injector and an S3115 conductivity detector (both SYKAM
Chromatographie Vertriebs GmbH, Germany), 175mg L-1 18-Crone-6 and 120µL
methanesulfonic acid served as the eluent with a set flow rate of 1.2mL $min^{-1}$. The injection
volume was 50µL. The column oven temperature was set at 30°C. The Cation Multi-Element
IC-standard (Carl Roth GmbH + Co. KG, Germany) containing $NH_4^+$, $Ca^{2+}$, $K^+$, $Li^+$, $Mg^{2+}$, $Na^+$
was measured before every replication series. For anions, the IC system consisted of a SeQuant
SAMS anion IC suppressor (Merck KGaA, Germany), an S5200 sample injector, a 3.0 x
150mm Sykrogel A 01 column and an S3115 conductivity detector (all SYKAM
Chromatographie Vertriebs GmbH, Germany). 6mM $Na_2CO_3$ with 90µM sodium thiocyanate
served as the eluent with a set flow rate of 1 ml $min^{-1}$ and a column oven temperature of 50°C.
The injection volume was 50 µL. The multi-element anion standard containing $F^-$, $Cl^-$, $Br^-$, $NO_2^-$
, $NO_3^-$, $PO_4^{3-}$ and $SO_4^{2-}$ was measured before every replication series. The standards and
samples were measured in triplicates.

### 272 2.3.4 Weathering indices and pedogenic oxide ratios

The KN Index A $(SiO_2+CaO+K_2O+Na_2O)/(Al_2O_3+SiO_2+CaO+K_2O+Na_2O)$ was calculated
after Kronberg and Nesbitt (1981). The index is based on the relative enrichment of the Al and
Si oxide phase and the leaching of Na, K and Ca. It ranges between 0 (prevailing chemical
weathering) and 1 (prevailing physical weathering). To get more precise information on the

ongoing chemical weathering, the chemical index of alteration (CIA) [($Al_2O_3$/($Al_2O_3+Na_2O+CaO^*+K_2O$)) x 100] after Nesbitt and Young (1982), in which CaO* represents the amount of silicate-bound CaO, was calculated. The CIA is frequently used as a quantitative measure of feldspar breakdown, assuming that feldspar represents the most abundant and reactive mineral. Higher values indicate increasing weathering intensity. Additionally, the degree of iron release ($Fe_d/Fe_t$) after Blume and Schwertmann (1969) was calculated, which gives information on the iron release from primary Fe-bearing mineral weathering: a longer or more intensive weathering process is indicated by a higher ratio (Baumann et al., 2014; Mirabella and Carnicelli, 1992).

**2.4 Micromorphology**

Samples for thin section preparation were air dried and afterwards embedded with a mixture of resin (Viscovoss N55 S, Vosschemie, Germany), stabilized Styrene (Merck KGaA, Germany) and hardener (MEKP 505 F, Vosschemie, Germany). After hardening, the samples were formatted into plane-parallel blocks and halved in the middle using a saw (Woco Top 250 A1, Uniprec Maschinenbau GmbH, Germany), and then one half was ground with the grinding machine (MPS-RC Vacuum, G&N GmbH, Germany) and mounted onto a glass carrier. Then the mounted samples was sawed into slices of about 150μm thickness. Finally, these slices were ground to a thickness of 25μm. The preparation followed the instructions given by Kühn et al. (2017). Afterwards, they were analyzed by using a polarizing microscope (ZEISS Axio Imager.A2m, Software AxioVision 4.7.2, Carl Zeiss Microscopy GmbH, Germany) and described following the terminology of Stoops (2003).

**2.5 Microbial community analysis**

**2.5.1 Nucleic acids extraction**

For each soil sample (maximum amount of 0.5g per sample), triplicates of total genomic DNA were extracted using the FastDNA™ Spin Kit for Soil (MO BIO Laboratories Inc., USA). The extracted DNA was stored at -20°C and used as a template for the enumeration of target genes by quantitative PCR (qPCR) and next-generation sequencing (Illumina HiSeq).

**2.5.2 Quantification of bacterial 16S rRNA gene copy numbers**

qPCR was used to quantify total bacterial abundances. All qPCR assays were performed in triplicates on a CFX96 Real-time thermal cycler (Bio-Rad Laboratories Inc., CA, USA) and contained 10μl SensiFAST SYBR Mix (Bioline GmbH, Germany), 5.92μl PCR water, 0.04μl

of forward and reverse primer (100μM) and 4μl template. The quantification of the bacterial 16S rRNA gene was based on the primers 341F (5'-CCTACGGGAGGCAGCAG-3') and 534R (5'-ATTACCGCGGCTGCTGG-3') according to Muyzer et al., 1993. After an initial denaturing phase of 3 minutes at 95°C, the cycler included 35 cycles of 3 seconds at 95°C, 20 seconds at 60°C and 60 seconds at 72°C plus the plate read. All cycling programs included a melting curve from 60°C to 95°C with 0.5°C steps per plate read. The analysis of quantification data was performed with the CFX Manager™ Software (Bio-Rad Laboratories Inc., CA, USA).

### 2.5.3 Illumina HiSeq-Sequencing

Unique combinations of tagged 515F (5'-GTGCCAGCMGCCGCGGTAA-3') and 806R (5'-GGACTACHVGGGTWTCTAAT-3') (Caporaso et al., 2010) primers were assigned to each sample (Tab. S1, S2). For each sample, two PCR reactions were prepared and the PCR product pooled after PCR reduce PCR variability. The PCR was performed on a T100™ Thermal Cycler (Bio-Rad Laboratories Inc., CA, USA) in 25μl reactions, containing 0.125μl OptiTaq DNA Polymerase and 2.5 10x Pol Buffer B (Roboklon GmbH, Germany), 1μl $MgCl_2$ (25mM), 1μl dNTP Mix (5mM), 16.625μl PCR water, each 0.625μl of forward and reverse primer (20μM) and 2.5μl genomic DNA. The following cycler program was used: Initial denaturing step for 3 minutes at 95°C followed by 10 cycles of 1 minute at 94°C, 1 minute at 53°C (-0.2°C/cycle) and 1 minute at 72°C, followed by 20 cycles of 1 minute at 94°C, 1 minute at 50°C and 1 minute at 72°C, followed by a final extension step for 10 minutes at 72°C. All barcoded samples were pooled into a single sequencing library by adding an equal amount of DNA (60ng DNA per sample). Subsequently, a purification of the PCR product pool was achieved by using the Agencourt AMPure XP – PCR Purification (Beckman Coulter, Inc., CA, USA). The Illumina HiSeq-sequencing was performed by GATC Biotech AG, Germany.

### 2.5.4 Bioinformatics and statistical analysis

Sequencing was performed on an Illumina HiSeq (2 x 300 bp). Dual-indexed reads were demultiplexed using CutAdapt (options: e0.1; trim-n; Martin, 2011). Barcode base pairs were required to have a phred quality score of Q25 and no mismatches were allowed. Read pairs were merged using PEAR (options: Q25; p10$^{-4}$; o20; Zhang et al., 2013). The orientation of all sequences were standardized by an own script using the information from demultiplexing. Sequences containing low-quality base pairs were trimmed and filtered using Trimmomatic (quality score of at least Q25 for trailing and leading base pairs, sliding window length of 5 basepairs, minimum sequence length of 200; Bolger et al. 2014). QIIME (version 1.9.1)

(Caporaso et al., 2010) was employed for microbiome analysis. USEARCH 6.1 (Edgar, 2010)
was used for the detection and removal of chimeric sequences. The SILVA database (version
128) (DeSantis et al., 2006) was utilized for the clustering of operational taxonomic units
(OTUs) (97% sequence similarity) and their taxonomic assignments. Singletons, OTUs
assigned to chloroplasts and mitochondria as well as rare OTUs (relative abundance of <0.1%
within each sample) were removed. Sample triplicates were merged by the mean value of their
relative abundance before visualization of the sequencing data and before analysis of correlating
environmental factors. For the processing and visualization of the obtained OTU table, R and
PAST3 (Hammer et al., 2001) were used. The hierarchical clustering of the samples using the
average linkage method was based on the Bray-Curtis dissimilarity. CANOCO5 (Šmilauer and
Lepš, 2014) was used for the canonical correlation analysis (CCA). If the Bonferroni corrected
*p*-value was <0.05, a given environmental parameter was included. Demultiplexed raw
sequencing data were submitted to the European Nucleotide Archive
(http://www.ebi.ac.uk/ena) under accession number: PRJEB29853.

## 3 Results

### 3.1 Field properties and soil classification

Both soils derived from coarse-grained marine sand- and siltstones, which were covered with
volcanic clasts. There was a higher contribution of volcanic material in BB than in SMC. The
amount of coarse material > 2mm was larger at the profile BB. Deflation processes led to a
residual enrichment of larger grains and pebbles at the soil surface of both profiles. The
permafrost table was not reached in both soil profiles, but ground ice was visible in a depth of
85cm at SMC. Neither SMC nor BB showed any ornithogenic influence. Both sites were
unvegetated by cryptogamic or vascular plants. The C-horizon was the only distinct soil horizon
occuring at SMC, whereas BB shows two changes within horizontal structures by abrupt
textural change below 10 cm and 20 cm. The textural change below 20 cm goes along with a
change in textural class; SCL (Sand: 52.5%, Silt: 21.9% and Clay: 25.6%) - CL (Sand: 44%,
Silt: 27.2% and Clay: 28.8%). Different from macroscopic features of the soil profiles, both
soils showed evidences of a downward transport and accumulation of particles and nutrients,
e.g. soluble products most likely originating from sea spray (Tab. 1). Accumulation starts at a
depth of 50cm at SMC and below 20cm at BB. Soil color did not change through the profiles.
SMC was brown to yellowish brown and BB was brownish yellow.

Both soils were classified as Cryosols (eutric, loamic) according to the WRB (IUSS Working Group WRB, 2015).

**3.2 Grain size distribution and soil chemistry**

SMC had higher sand contents (mean value 61.7%, Table 1), while BB was characterized by lower sand contents (mean value 47.4%) and higher silt and clay contents (mean values 25.3% and 27.2% respectively). The grain size distribution varied only slightly with depth and similar clay and silt contents were demonstrated for both soils.

The pH was slightly to moderately alkaline in both profiles and highly alkaline only in the upper 5cm of BB. The pH values followed opposing trends with depth, increasing in SMC from 7.7 to 8.1 and decreasing in BB from 8.6 to 7.4. The EC ranged between 50-60µS cm$^{-1}$ in SMC and was substantially higher in BB with a minimum of 350-450µS cm$^{-1}$ within 5-50cm and its highest values around 900µS cm$^{-1}$ between 0-5cm and from 50cm downwards. According to the EC values, SMC and the middle part of BB can be considered as being salt-free, whereas the salt content in the upper and lowermost part of BB was low (Food and Agriculture Organization of the United Nations (FAO), 2006).

The total inorganic carbon (TIC) content was low in both soils ranging between 0.1 and 0.3mg g$^{-1}$ in SMC and between 0.7 and 2.0mg g$^{-1}$ in BB. The TOC content ranges from 0.8-0.9mg g$^{-1}$ for SMC and from 1.4 and 2.6mg g$^{-1}$ for BB and increased there slightly with depth. The N content was around 0.4mg g$^{-1}$ across both soil profiles. The C/N ratio was generally low with values below 7.5 in both soils, it decreased with depth in SMC (2.6 – 2.1) and increased with depth in BB (4.0-7.4).

Ion concentrations (Tab. 1) were parallel to the depth function of the conductivity in both soils; e.g. higher EC and ion concentration characterized BB. Cl$^-$ concentrations decreased with depth in SMC from 20.5 to 3.5µmol g$^{-1}$ soil as well as in BB from 4,522 to 231µmol g$^{-1}$ soil. The highest SO$_4^{2-}$ concentrations were observed in the shallow (SMC: 9.6µmol g$^{-1}$ soil; BB: 621µmol g$^{-1}$ soil) and deepest (SMC: 15.3µmol g$^{-1}$ soil; BB: 451µmol g$^{-1}$ soil) samples. K$^+$, Mg$^+$ and Ca$^+$ concentrations followed the same trend as SO$_4^{2-}$. Br$^-$, NO$_2^-$, NO$_3^-$ and PO$_4^{3-}$. Li$^+$ and NH$_4^+$ concentrations were below the detection limit.

**[ Table 1 ]**

**3.3 Weathering indices and pedogenic oxide ratios**

Weathering indices were calculated according to the major element contents (Table 3). The KN Index A was at 0.91-0.92 in SMC and only slightly lower with 0.89 - 0.90 in BB (Table 2). The CIA varied between 53.9 and 54.8 in SMC and between 56.9 and 58.8 in BB. Both indices indicated weak chemical weathering with a slightly higher weathering intensity in BB.

**[ Table 2 ]**

The $Fe_d/Fe_t$ ratio showed a decreasing trend from 0.18 to 0.11 with depth in SMC indicating a decreasing intensity of pedogenic processes with depth. No particular trend was found in BB; but the $Fe_d/Fe_t$ ratio is – similar to the CIA - generally higher around 0.20 except for 0.16 in the upper 5cm.

**3.4 Micromorphology**

Formation of platy and lenticular aggregates due to repeated freezing and thawing processes was detected. Neither platy and lenticular platy structures nor the results of translocation (eluviation) processes were observed during fieldwork, but could be confirmed later using micromorphology.

SMC had a weak to moderately developed pedality and a weak to moderate degree of separation (Table 3). Both, pedality and degree of separation are well developed at a depth of 50-60cm and were lowest developed close to the surface and at the bottom of the profile. In contrast, BB had a well-developed pedality and a moderate to high degree of separation with its maximum development close to the bottom of the profile.

**[ Table 3 ]**

Lenticular and subangular blocky microstructures were present in both profiles, whereas lenticular microstructure was dominant in SMC and subangular blocky microstructure was dominant in BB. Lenticular shaped aggregates were first observed at a depth of 10cm in profile BB, and at 30cm in SMC (Figures 3a and 3b).

**[ Figure 3 ]**

Translocations features, like cappings consisting of clay and silt particles welded together with sand-sized quartz grains were present in the upper part of both profiles. Link cappings occurred

in the lower part of both profiles, with lesser and smaller cappings in BB (Fig. 3d). Link cappings were very rare and occurred only where coarse rock fragments were located close to each other. Dusty silt and clay pendants occurred only in the lower part of BB (20-50cm) (Fig. 3e). The sphericity of mineral grains was smooth in both profiles. The minerals were slightly better rounded in BB (subangular to round) than in SMC (subangular to subrounded). Weathering processes were identified by pellicular and dotted alteration patterns on rock fragments (mostly in sandstone fragments) in both profiles with a higher number of fragments with dotted alteration patterns than with pellicular alteration patterns. The quantity and intensity of dotted alteration patterns decreased with depth. Larger rock fragments were often strongly weathered, so that mainly quartz-minerals were still preserved (Fig. 3f). Besides quartz, glauconite is the main mineral component in the unweathered sandstone fragments. In addition, feldspars and micas occur to a very small extent. The sandstones cemented by fine material and faint Fe coatings are visible around quartz grains. Pellicular alteration pattern was found exclusively on volcanic rock fragments, and only in the uppermost thin section (0-10cm) of SMC (Fig. 3g). Fragments showing pellicular alteration patterns occurred in 10-30cm of BB. Even though the number of weathered fragments decreased, pellicular patterns were slightly thicker in slide BBII (20-30cm) than in BBI (10-20cm). However, pellicular alteration patterns did not exceed the state of "pellicular" in any analyzed slide whereas dotted alteration patterns often reach the state of "patchy cavernous residue" (Fig. 3e) and do occur also as dispersed minute residues (Stoops, 2003).

## 3.5 Microbial abundance and community structure

The enumeration of the 16S rRNA gene revealed a similar trend for both soil profiles (Fig. 4). The highest abundances with $6.6 \times 10^8$ copies $g^{-1}$ soil (BB) and $1.7 \times 10^8$ copies $g^{-1}$ soil (SMC) were detected in the uppermost depth increment of both soil profiles. Both soils showed a decrease in bacterial abundances with depth. The lowest bacterial abundances in SMC were detected below 50cm depth with $3.7 \times 10^5$ copies $g^{-1}$ soil, and in BB in 20-50cm depth with $1.7 \times 10^6$ copies $g^{-1}$ soil.

In total, 19,732,536 reads were obtained after merging the forward and reverse reads, demultiplexing, filtering, and deletion of chimeric and singleton sequences. Additionally, reads of chloroplast-associated OTUs (36,573), mitochondria-associated OTUs (1,117) as well as rare OTUs (OTUs with a relative abundance of <0.1% in every sample; 4,287,382) were filtered, resulting in 15,407,464 reads (Tab. S4). The number of reads per sample ranged from 54,122 to 916,583 with a mean value of 513,582. A total of 687 OTUs was clustered. After taxonomic

classification, 258 putative taxa were obtained. Shannon's H index was used to estimate and compare the alpha diversity of the different depth increments interval of the soils (Tab. S5). Both soils showed a similar Shannon's H index, which ranged from 3.7 to 4.7 not following any specific trend.

Bacteria dominated the microbial community in both soil profiles (Fig. 4). Higher abundances of Thaumarchaeota (7.2 - 12.9%) were found in the upper 10cm of the soil profile from SMC (Tab. S4). On a phylum level, the soil profile of SMC was dominated by Proteobacteria (23.4 - 57.9%) and Actinobacteria (17.7–41.3%) but showed also relative high abundances of Acidobacteria (3.9-14.1%). The microbial community in BB was also mainly composed of Proteobacteria (28.2-30.8%), followed by Actinobacteria (27.6-46.6%), Gemmatimonadetes (3.9-24.7%) and Chloroflexi (5.3-10.9%). Bacteroidetes were highly abundant (10.5%) in the top 5 cm of BB. Regarding potential phototrophic organisms in the investigated soils, the amount of chloroplast-related reads was relatively low (<0.2%) in each sample, except for SMC >50 cm (0.03% - 1.30%) and BB 0 – 5 cm (0.87% - 1.01%). Cyanobacteria-related OTUs were rare and only showed low relative abundances in SMC 5 – 10cm (0.06%), SMC 10 – 20cm (; 0.62%), SMC >50cm (0.04%).

**[ Figure 4 ]**

The distribution of dominant OTUs was reflected by a cluster analysis based on the Bray-Curtis dissimilarity of the investigated depth increments. Samples were clustered according to their origin and depth. On a first level, samples grouped according to depth in upper (0–20cm) and deeper (20-80cm) samples and within these groups they clustered according to location (BB vs. SMC). An exception is the sample from BB from the depths 0–5cm which formed an own cluster (Fig. 5). The deeper samples in both profiles (20–80cm depth) showed high relative abundances of three OTUs related to Acidiferrobacteraceae(1, 2, 3) (SMC: 1.7-14.6%; BB: 2.2-9.8%) and one OTU related to Gemmatimonadaceae(1) (SMC: 1.5-3.8%; BB: 14.1-20.3%). High proportions of two OTUs related to Gammaproteobacteria(1, 2) (SMC: 2.8-11.4%; BB: 5.4-10.2%) and one OTU related to Gaiellales(2) (SMC: 3.7-5.7%; BB: 7.2-8.3%) were observed in the shallow samples (0-20 cm). BB 0-5 cm was comprised of a strongly different community. The most abundant taxa in this sample were related to *Thermomonas*(1) (6.4%), *Sphingomonas* (3.7%) and Solirubrobacterales(1) (3.7%).

**[ Figure 5 ]**

The relationship of OTU distribution and environmental parameters was examined by applying
a CCA (Fig. 6). Contents of chloride (18.5%), calcium (11.8%), sulfate (5.9%), silt (5.6%),
TOC (6%) and the $Fe_d/Fe_t$-ratio (12.5%) formed the optimal subset to explain variations in
community structure of the investigated soil profiles ($p$ <0.05). The adjusted explained
compositional variation was 49.9%. A strong correlation between the unique community of BB
0-5cm and the saline conditions was observed, mainly caused by high sulfate and chloride
concentrations. The remaining samples were arranged according to sample site and depth as
already observed in the cluster analysis above.

**[ Figure 6 ]**
**4 Discussion**
The interaction of biotic and abiotic processes remains one of the fundamental questions in
ecosystem research and further the initial development of soils under harsh environmental
conditions, such as Antarctica. So far, only a few studies exist for polar environments that
integrate pedogenic and microbiological research (e.g. Aislabie et al. 2008, Cowan et al. 2014,
Ganzert et al. 2011; Bajerski and Wagner, 2013). James Ross Island offers an exceptional
opportunity to improve our understanding of the interrelations between soil formation and
microbiological properties in the absence of plants. The present interdisciplinary study gives
profound insights in the state of soil formation and microbial community structure in initial
soils in the transition zone between maritime and continental Antarctica.
James Ross Islands is located in the transition zone between warmer and wetter maritime
Antarctica and cold and dry continental Antarctica (Souza et al., 2014). In this area, we studied
two representative soils 16km apart, with different exposures to the dominant south-westerly
winds. The leeward position of SMC displays formation conditions of a typical inland soil,
while BB in its windward position represents coastal soils. As indicated by EC values, BB is
influenced by sea spray, while SMC, sheltered behind the Lachman Crags, does not show strong
input of soluble salts from sea spray.
The examined soils on JRI were characterized by low TOC (0.9-2.6mg g$^{-1}$) and low total
nitrogen contents (approx. 0.4mg g$^{-1}$), which is common for Antarctic soil environments (e.g.
Cannone et al., 2008), and relative high pH values (7.4- 8.6). The moderately to highly alkaline
pH in both soils cannot be explained by the occurrence of $CaCO_3$, because the soils have a
negligible amount with of TIC ($\leq$ 2mg g$^{-1}$). Low C contents do not only show the missing
influence of penguins, but also indicate a relative juvenility of the soils: This indicates that no
cations have been leached from the topsoil, and therefore the pH remains neutral to basic
(Wilhelm et al., 2016). In addition, the content of basalt clasts in the parent material results in
increased soil pH values (Simas et al., 2002; Moura et al., 2012). The opposing trends in the
depth function of the pH values are caused by the input of soluble salts from sea spray: wind
can transport soluble salts from the sea causing an additional input of bases simultaneously
increasing the pH at BB, while SMC is not affected (Benassai et al., 2005; Russell et al., 2010;
Hara et al., 2004; Udisti et al., 2012). Since the substrate was not colonized by plants, lichens
or endolithic prokaryotes, and the taxonomic data revealed low abundances of phototrophic
organisms, the alkalization of the substrate by the release of hydroxyl ions in the course of
photosynthesis has a minor effect on soil pH. On the other hand, the neutral to basic pH does
not significantly affect the soil microbial community structure, which is in accordance with
observations in soils from Livingston Island (South Shetland Archipelago, maritime Antarctica)
by Ganzert et al. (2011). They explained it by the occurrence of a specific soil microbial
community, which thrives under low C and N conditions and is not depending on nutrient input.
Therefore, pH is mainly driven by the parent material composition combined with the input of
soluble salts in these young soils on JRI.
The additional input of airborne cations by sea spray led to higher sodium and calcium contents
and a rejuvenation of the affected depth increments of the soil profile, which can be seen in the
lower CIA values in 0-5 cm soil depth of both soils compared to the lower part of the profiles.
Ions, for instance sulfate accumulate close to the permafrost table, which acts as a barrier and
therefore explains increasing contents of sulfate with depth. The high amount of sulfate near
the surface is most likely caused by sea spray and precipitation, because they are known to carry
high amounts of sulfate in coastal areas (Blume et al., 2010).
Chemical weathering, as indicated by the KN-Index A (Kronberg and Nesbitt, 1981), is only of
minor importance whereas physical weathering is prevailing. The CIA and pedogenic oxide
ratios (POR) confirmed the low degree of soil formation. Pedogenic oxides with specific
degrees of crystallization relate to intensity and/or duration of pedogenic processes (Baumann
et al., 2014; Blume and Schwertmann, 1969; Mirabella and Carnicelli, 1992). The results show
that both CIA and both POR are slightly higher at BB compared to SMC. The KN-Index A and
the CIA showed a weak chemical weathering of these mineral soils (Michel et al., 2014). Both
indices indicated a more intensive chemical weathering at BB and, thus, indicate a slightly
stronger pedogenesis at BB than at SMC. This finding could be explained by the sea- and
windward position of BB, which results in an increased water availability and a slightly more
levelled microclimate. Since both soils are located in similar topographic positions and derived

from similar parent material, CIA and POR results allow the interpretation that soils influenced by coastal conditions tend to be more weathered. Besides physical and chemical weathering, microorganisms play an important role in mineral dissolution and oxidation. Adapted microorganisms colonize minerals and are, depending on nutritional requirements, nutrient availability and mineral type, potential contributors to the weathering of minerals (Uroz et al., 2009). Taxonomical groups, which are usually connected to microbial weathering, are present in the soils, such as *Massilia, Bacillus* (Ma et al., 2011) and *Polaromonas* (Frey et al., 2010). Interestingly, the relative abundances of these taxa changed according to the degree of weathering. This could indicate a possible interrelation between the occurrence of these potential weathering-related organisms and the degree of weathering of Antarctic soils.

Evaluating weathering using the CIA, it must be noted that the value for BB is most likely underestimated. Ion chromatography results show that Na-content is much higher at BB. The high amount of Na is most likely caused by sea spray, which is known to carry high amounts of Na (Udisti et al., 2012). Since the calculation of the CIA takes Na into account (Nesbitt & Young, 1982), the CIA values would be significantly higher if the additional input of sea salts could be excluded. It is very likely that the actual difference in state of weathering between SMC and BB would be much higher. In conclusion, chemical weathering, even without influence of guano deposits, is of higher importance for the current state of soil formation, than the ongoing cryoturbation.

In case of the pedogenic oxide ratios, 12.5% of the total compositional variation could be explained, which indicates a correlation between the microbial community structure and weathering at a very initial stage of soil formation. The pedogenic oxide ratios correlate with the compositional distribution of microorganisms in the investigated soils, and with the relative abundances of one Acidiferrobacteraceae-related OTU. Microorganisms of this family are described as autotrophic sulfur and iron oxidizers, which have the capacity to use ferrous iron, thiosulfate, tetrathionate, sulfide and elemental sulfur as electron donors and oxygen or ferric iron as terminal electron acceptor (Hallberg et al., 2011). The reactive iron could potentially be used as terminal electron acceptor in the course of microbial iron cycling (Canfield, 1989). Organic matter, a potential substrate for heterotrophic microbial processes, sorbs on mineral surfaces (Kaiser and Guggenberger, 2000) and could be released in the course microbial oxidation and reduction of reactive iron phases. In addition to the autotrophic processes, the release of sorbed, organic matter from mineral surfaces could be an additional way to increase the pool of biologically available carbon. The availability of such a mechanism potentially has

an influence on the microbial community structure and abundances in oligotrophic environments.

Translocation features are common features in permafrost-affected soils. They often occur together with platy rectangular or lenticular aggregates, caused by reoccurring freeze-thaw-cycles (Van Vliet-Lanoë, 1985). Platy blocks and lenses dominated the microstructure in the areas between 20 and 50cm of both profiles. They were absent near the surface of both profiles and at the bottom of the profile SMC. These microstructures are known to occur in the transition zone between permanently frozen and unfrozen soils (Shur et al., 2005; Van Vliet-Lanoë et al., 2004). Here, the alternating temperature and soil moisture conditions additionally affect the microbial community structure. The frequency of freeze-and-thaw cycles tends to be steady in the middle part of a permafrost-affected soil, whereas weather shifts influence the surface, causing several freeze-and-thaw events per day, which do not result in typical microstructure formation due to insufficient water supply (Van Vliet-Lanoë, 1985). Aggregate formation by reoccurring freeze-and-thaw cycles result in a change in pore shape and size (Van Vliet-Lanoe et al., 2004). Especially during the summer season, intensive insolation causes high evaporation, resulting in dry soil surfaces. Changes in pore space affects microbial habitats, due to larger pores and a more sufficient water supply. This has a severe influence on matter fluxes and soil-environmental conditions, which is reflected in a changing species distribution and, more specifically, the occurrence of different clusters of highly abundant organisms in both soils. Multivariate statistics were performed for soil depth increments considered as being independent. However, when processes are discussed that link between soil horizons, e.g. water and solute flow through the profiles, we account for the limited number of two soil profiles with great care. We could not detect any environmental factors that increase or decrease the correlation between the chosen depth increments. Nevertheless, freeze-and-thaw cycles definitely also occur in the upper part of the profile, as indicated by the well sorted areas (Van Vliet-Lanoë, 1985), which were described as single grain microstructure. Near the permafrost table aggregates are often formed by frost desiccation and are hence poorly compacted what makes them unstable upon moistening, which occurs during thawing events and explains the missing platy microstructure at SMC near the bottom of the profile (Van Vliet-Lanoë, 2010). The fact that lenticular shaped aggregates occur also in the lower part of the profile indicates that the permafrost table is located underneath the layer of coarse gravel at BB.

Although the investigated soils were poorly developed, an abundant and diverse prokaryotic community could be observed. Microbial abundances in both soils showed a decreasing trend with depth. Values of up to $10^9$ gene copies $g^{-1}$ soil in the uppermost depth increments are

comparable to observed microbial abundances from other cold environments, such as alpine glacial forelands (Sigler et al., 2002), permafrost-affected soils from arctic regions (Liebner et al., 2008) and Antarctic glacier forefields (Bajerski and Wagner, 2013).

Both soils were characterized by a highly diverse community dominated by Proteobacteria, Actinobacteria, Gemmatimonadetes, Acidobacteria and Chloroflexi, which is in accordance with the observations in other continental and maritime Antarctic habitats (e.g. Yergeau et al., 2007; Cary et al., 2010, Ganzert et al., 2011, Bajerski and Wagner 2013, Wang et al., 2016). Substantial differences in geochemical parameters such as conductivity, the change of the community structure on a phylum level were evident as well as the occurrence of depth-dependent clusters (0-20 cm; >20 cm) of dominant OTUs (Fig. 8). Whereas the upper 20cm of the soils were dominated by Gammaproteobacteria and Gaiellales, the deeper part of the soils showed increased abundances of OTUs related to Acidiferrobacteraceae and Gemmatimonadaceae. This distinct shift correlates with the occurrence of the microstructure related to freezing and thawing and could be related to its changes of the pore space and the availability of oxygen, water and nutrients. For instance, Gemmatimonadaceae were a common observation in the soils and showed increased abundances in deeper parts of BB. These organisms have a cosmopolitan distribution in terrestrial environments and depend on the soil moisture condition of the respective soil and soil depth (DeBruyn et al., 2011; Bajerski and Wagner, 2013). Only a few isolates have been described for this phylum (e.g. Zeng et al., 2015) and their exact functions in soil ecosystems remain uncertain. The change in relative abundance of these taxa with depth could be coupled to the changing availability of water, which depends on the microstructure. For example, the amount and size of microaggregates have been shown to be important regarding prokaryotic colonization, leading to genetically distinct communities as well as cell densities in different size classes of aggregates (Ranjard et al., 2000). Thus, in addition to environmental parameters, which shape the overall prokaryotic community, the microstructure of the initial soils could have a substantial influence on species distribution.

Higher abundances of Bacteroidetes- and especially Flavobacteriaceae-related OTUs were observed in the uppermost area of soil from BB, while only showing minor abundances in the deeper soil areas. This area differed from the remaining soil in two regards, namely very high chloride concentrations and a relative high content of coarse sandy material and could select for adapted psychro- and halotolerant Bacteroidetes-related organisms, such as Flavobacteriaceae (e.g. Bajerski et al., 2013a). Members of the Flavobacteriaceae family detected in this area, for instance *Gillisia sp.*, were isolated from Antarctic habitats before and were shown to be at least moderately tolerant to saline conditions (Bowman and Nichols, 2005).

Putative halotolerant or halophilic Flavobacteriaceae in this area could have a need for high chloride contents. Chloride can be accumulated inside the cell to osmotically balance the cytoplasm with the surrounding habitat (Oren et al., 2002; Müller and Oren, 2003). Furthermore, the detected Bacteroidetes-related organisms could prefer the coarser, sandy microstructure from this depth increment. The preference of microbial groups for certain grain-size-dependent microenvironments, for instance the sand-sized fraction being preferred by Bacteroidetes, was shown, e.g. in Typic Hapludalfs from central Denmark (Hemkemeyer et al., 2018).

Both investigated soils were poor in soil organic C as well as N. Organisms with the ability to use oxygenic photosynthesis to fixate $CO_2$, such as cyanobacteria, were nearly absent in the investigated soils. Low abundances of Cyanobacteria are a common observation for Antarctic soil habitats (Ji et al., 2016). Due to the lack of phototrophic organisms and organic carbon, inorganic compounds and metabolic pathways utilizing those may have a more pronounced role in sustaining the microbial ecosystem at this initial stage of the soils. Several of the most abundant taxa observed in BB and SMC were putative chemoautotrophs involved in nitrogen, iron and sulfur cycling, such as potential ammonia-oxidizing Thaumarchaeota or sulfur/iron-oxidizing Acidiferrobacteraceae. Microorganisms can be seen as the primary pioneers of nutrient-poor environments such as Antarctic soils, and were shown to have the genetic potential to fix C and N (Cowan et al., 2011; Niederberger et al., 2015), thus increasing C and N contents of these oligotrophic soils. The chemoautotrophic Thaumarchaeota oxidize ammonia aerobically to nitrite (Brochier-Armanet et al., 2008; Vajrala et al., 2013) and were observed in many studies located in Antarctica (Magalhães et al., 2014; Ayton et al., 2010). However, ion chromatography showed that amounts of ammonia as well as nitrite and nitrate were negligible in both soils. Ammonia originating from necromass and products in the course of nitrification could be metabolized directly by the present community, so no accumulation of the different intermediates containing nitrogen takes place. These organisms are reported to have the genetic potential to use the hydroxypropionate/hydroxybutyrate pathway for $CO_2$ fixation, which is highly efficient and could provide an ecological advantage in oligotrophic environments (Könneke et al., 2014). Further, a part of the community could use atmospheric compounds as energy source. Atmospheric $H_2$, CO, and $CO_2$ are scavenged and used as an energy source by microorganisms, especially organisms associated with the phyla Actinobacteria, Chloroflexi, Acidobacteria, Planctomycetes, Verrucomicrobia, and Proteobacteria (Greening et al., 2015; Ji et al., 2017) .Operational taxonomic units related to the phylum Actinobacteria and the associated orders Acidimicrobiales and Solirubrobacterales

were highly abundant in the investigated soils. Microorganisms in Antarctic soils, especially bacteria related to the phyla Actinobacteria, AD3 and WPS-2, were shown to generate biomass by consuming $H_2$, $CO_2$ and CO from the atmosphere (Ji et al., 2017). The gene for chemosynthetic $CO_2$ fixation, *rbcL1E*, was found in multiple orders, including Pseudonocardiales, Acidimicrobiales and Solirubrobacterales. Similar functional capabilities could be present and active in the investigated soils. Our results show that, in this initial stage of soil development, chemolithoautotrophic lifestyles plays an important role for the generation of biomass and initial accumulation of soil organic carbon and nitrogen.

**5. Conclusion**

The presented soil and microbiological study on initial soils in the semiarid environment of Antarctica shows the current state of soil formation indicated by main soil and microbiological properties and their interplay. The results allow us to draw the following conclusions:

1. Despite similarities in topographic position and substrates, both profiles have distinct differences in chemistry (content of salts indicated by EC, opposing trends in pH and states of weathering, indicated by WI and POR) and microbiology (depth functions of microbial abundances and diversity, e.g. Proteobacteria, Gemmatimonadetes and Thaumarchaeota abundances), which are caused by the different local environmental conditions at each site.

2. The EC values of the soils and the depth function of the pH values clearly showed different conditions for soil formation at the two sites caused by the more exposed location of BB towards the mainly south-westerly winds, resulting in a more intense weathering and higher input of salt by sea spray.

3. Taking weathering and aggregation as indicators of soil formation, we conclude that coastal conditions - in contrast to inland conditions - favor the formation of soils in maritime Antarctica.

4. Despite different local environmental conditions at each site, the microbial communities differ more distinctly between the depth increments in one profile than between the two profiles. Therefore, we conclude that in this initial stage of soil formation factors such as weathering and microstructure formation, as well as the resulting parameters (e.g. water availability and matter fluxes), are more important drivers of soil microbial community composition than chemical parameters such as EC and pH.

5. Assuming that prokaryotic life is highly affected by changes in soil structure and vice versa, further investigations in this field should include analyses of (micro-) aggregates.

739

*Author Contributions.* The project was initiated and designed by Dirk Wagner, Peter Kühn, Thomas Scholten and Carsten W. Mueller. Lars A. Meier and Carsten W. Mueller carried out fieldwork during the PROANTAR fieldtrip led by Carlos E.G.R. Schaefer in 2016. Lars A. Meier, Patryk Krauze, Isabel Prater and Fabian Horn did analyses and interpretation. Lars A. Meier and Patryk Krauze prepared this manuscript with contributions from all co-authors.

*Competing interests.* The authors declare that they have no conflict of interests.

*Acknowledgements.* We thank the Brazilian Navy and the Brazilian Antarctic Expedition PROANTAR for all logistics and help in the field during southern summer 2015/2016. We especially acknowledge the supported by the German Research Foundation (DFG) in the framework of the priority programme 1158 'Antarctic Research with Comparative Investigations in Arctic Ice Areas' by a grant to DW (WA 1554/18), TS (SCHO 739/18), PK (KU 1946/8) and CWM (MU 3021/8).

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

## Tables

**Table 1: Soil properties of two soil profiles from St. Marta Cove (SMC) and Brandy Bay (BB) from James Ross Island, Antarctica.**

| Sample | Depth [cm] | $pH_{H2O}$ | EC [µS cm$^{-1}$] | TIC [mg g$^{-1}$] | TOC [mg g$^{-1}$] | N [mg g$^{-1}$] | C/N | Na$^+$ [µmol g$^{-1}$] | K$^+$ [µmol g$^{-1}$] | Mg$^+$ [µmol g$^{-1}$] | Ca$^+$ [µmol g$^{-1}$] | Cl$^-$ [µmol g$^{-1}$] | SO$_4^{2-}$ [µmol g$^{-1}$] | Sand 63–2000 µm [%] | Silt 2-63 µm [%] | Clay <2 µm [%] |
|---|---|---|---|---|---|---|---|---|---|---|---|---|---|---|---|---|
| SMC 0-5 | 0-5 | 7.7 | 46 | 0.1 | 0.9 | 0.4 | 2.6 | 41.5 | 2.5 | 4 | 10.4 | 20.6 | 9.6 | 61.2 | 18.9 | 19.8 |
| SMC 5-10 | 5-10 | 8 | 36 | 0.1 | 0.9 | 0.4 | 2.5 | 30.4 | 2.4 | 3.6 | 9.6 | 13.1 | 5.7 | 59.9 | 19.4 | 20.7 |
| SMC 10-20 | 10-20 | 7.9 | 33 | 0.3 | 0.9 | 0.4 | 2.3 | 27.1 | 2 | 3.1 | 8.3 | 8.7 | 3.3 | 63.8 | 17.1 | 19.1 |
| SMC 20-50 | 20-50 | 8 | 33 | 0.1 | 0.8 | 0.4 | 2.2 | 38.6 | 1.5 | 2.1 | 4.9 | 5.5 | 3 | 61.9 | 17.2 | 20.8 |
| SMC > 50 | >50 | 8.1 | 65 | 0.2 | 0.9 | 0.4 | 2.1 | 91.5 | 2.7 | 3.1 | 6.3 | 3.5 | 15.3 | 61.7 | 20 | 18.3 |
| BB 0-5 | 0-5 | 8.6 | 950 | 1.4 | 1.4 | 0.4 | 4 | 1590 | 23.4 | 84.6 | 151 | 4522 | 621 | 49.8 | 25.2 | 24.9 |
| BB 5-10 | 5-10 | 8.1 | 561 | 1.2 | 2.1 | 0.4 | 5.6 | 470 | 16.3 | 57.4 | 108 | 702 | 123 | 46.4 | 25.7 | 27.9 |
| BB 10-20 | 10-20 | 7.7 | 385 | 0.7 | 2 | 0.3 | 5.9 | 268 | 12.2 | 42.6 | 93 | 369 | 88 | 52.5 | 21.9 | 25.6 |
| BB 20-50 | 20-50 | 7.6 | 505 | 2 | 2.5 | 0.4 | 6.7 | 191 | 18.3 | 79.8 | 173 | 386 | 163 | 44 | 27.2 | 28.8 |
| BB > 50 | >50 | 7.4 | 965 | 1 | 2.6 | 0.4 | 7.4 | 149 | 23.9 | 140 | 297 | 231 | 451 | 44.3 | 26.8 | 28.9 |

**Table 2: Weathering indices (WI) and pedogenic oxide ratios (POR) of two soil profiles from St. Marta Cove (SMC)**
**and Brandy Bay (BB) from James Ross Island, Antarctica. CIA = chemical index of alteration; KN-A = Kronberg**
**Nesbitt Index; Fe$_d$ = dithionite-soluble iron; Fe$_t$ = total iron; Fe$_o$ = oxalate-soluble iron.**

| Sample | Depth [cm] | WI | | | | POR | | |
|---|---|---|---|---|---|---|---|---|
| | | CIA | KN-A | Fe$_d$/Fe$_t$ | Fe$_o$/Fe$_d$ | Fe$_t$ [mg g$^{-1}$] | Fe$_d$ [mg g$^{-1}$] | Fe$_o$ [mg g$^{-1}$] |
| SMC 0-5 | 0-5 | 53.9 | 0.92 | 0.18 | 0.56 | 45.57 | 7.99 | 4.48 |
| SMC 5-10 | 5-10 | 54.2 | 0.91 | 0.18 | 0.45 | 44.71 | 7.83 | 3.56 |
| SMC 10-20 | 10-20 | 54.8 | 0.91 | 0.16 | 0.53 | 40.74 | 6.61 | 3.48 |
| SMC 20-50 | 20-50 | 54.3 | 0.91 | 0.15 | 0.59 | 40.76 | 5.96 | 3.53 |
| SMC > 50 | >50 | 54.1 | 0.92 | 0.11 | 1.72 | 42.25 | 4.83 | 8.3 |
| | | | | | | | | |
| BB 0-5 | 0-5 | 56.9 | 0.89 | 0.16 | 0.61 | 53.77 | 8.68 | 5.3 |
| BB 5-10 | 5-10 | 58.5 | 0.89 | 0.21 | 0.57 | 44.09 | 9.08 | 5.19 |
| BB 10-20 | 10-20 | 58.1 | 0.9 | 0.2 | 0.58 | 42.57 | 8.34 | 4.85 |
| BB 20-50 | 20-50 | 58.8 | 0.9 | 0.21 | 0.56 | 39.82 | 8.43 | 4.68 |
| BB > 50 | >50 | 58.2 | 0.9 | 0.21 | 0.54 | 38.18 | 7.88 | 4.24 |

**Table 3: Micromorphological features of two soil profiles from St. Marta Cove (SMC) and Brandy Bay (BB) from James Ross Island, Antarctica**

The micromorphological property is shown by the presence (cross) or absence (no cross). (x) = occasional occurring
* Microstructures separated by "/": two different microstructures were found. Microstructures separated by "()": one ms shows partly features of another ms
** Degree of roundness and sphericity results separated by "/": two different degrees were mainly present ; measured at 10x magnification.

| Slide | Depth [cm] | Aggregation Pedality wp | mp | hp | ds | Voids spv | xpv | pl | vu | Micros* | RS** | c/f-related distribution cm | cg | oee | ssee | chi | ce | Micromass color | b-Fabric u | gs | Pedofeatures Redoximorphic features nodules t | a | hp ro | li | Translocation features coatings cap | pen | infillings ld |
|---|---|---|---|---|---|---|---|---|---|---|---|---|---|---|---|---|---|---|---|---|---|---|---|---|---|---|---|
| SMC I | 0-10 | x | | | w | (x) | x | (x) | | fis / sgm | sub/su | x | (x) | | | (x) | | gb | x | | x | x | (x) | (x) | (x) | | |
| SMC II | 10-20 | | x | | w | | x | (x) | | pgm | su | (x) | x | | | x | | gb | x | | | x | | | x | | |
| SMC III | 30-40 | | (x) | | w/m | | x | x | | wsl | sub/su | x | x | | | x | (x) | db | x | x | x | | x | | x | | x |
| SMC IV | 50-60 | | x | | m/w | | x | x | (x) | msl (hsp) | sub/su | x | x | x | | | | db | x | x | | | | x | | | |
| SMC V | 80-90 | x | | | w | | x | (x) | | (fis) pgm | sub/su | x | (x) | | | x | x | db | x | (x) | | | | | (x) | | x |
| BB I | 10-20 | | (x) | | m | | x | x | (x) | h-m sp | sub/su | (x) | x | | x | (x) | | gb | x | (x) | x | | | | (x) | | x |
| BB II | 20-30 | | x | | m | | x | x | x | w-m sp (msl) | su-ro | (x) | x | | x | (x) | | gb | x | x | x | (x) | x | | x | x | x |
| BB III | 40-50 | | | x | m/h | | x | x | (x) | h-m sp (msl) | sub/ro | x | x | x | | x | | gb | x | x | x | x | x | x | x | x | x |

**Aggregation** : hp = highly developed pedality , mp = moderately developed pedality , wp = weakly developed pedality , w = weakly separated
ds = degree of separation: h = highly separated, m = moderately separated, w = weakly separated

**Voids** : spv = simple packing voids, xpv = complex packing voids, pl = planes, vu = vughs

**Microstructure *(Micros)** : fis = fissure, sgm = single grain ms, pgm = pellicular grain ms, wsl = weakly separated lenticular ms, hsp = highly separated platy ms
: msp = moderately separated platy ms, wsp = weakly separated platy ms, msl: moderately separated lenticular ms

**Groundmass**
RS - Degree of Roundness and Sphericity ** : sub = subrounded, su = subangular, ro = rounded, su-ro = subangular to rounded mineral grains
c/f - Related Distribution : cm = coarse monic, cg = chito-gefuric, oee = open equal enaulic, ssee = single spaced equal enaulic, chi = chitonic, ce = close enaulic
(c/f - R. Distr.)
color : gb = greyish brown, db = dark brown
b - Fabric : u = undifferentiated, gs = granostriated

**Pedofeatures**
nodules : t = typic, a = aggregate
hp (hypocoatings) : ro = redoximorphic hypocoatings
coatings : li = link cappings, cap = cappings, pen = pendent
infillings : ld = loose discontinuous

**Figures**

**Figure 1:** Mainmap shows the regional setting of Ulu-Peninsula on James Ross Island, Maritime Antarctica. Black circles indicate the location of both study sites, Brandy Bay (BB) and St. Marta Cove (SMC). Sidemap 1-3 provide an additional overview over Antarctica, the Antarctic Peninsula Region and James Ross Island.

**Figure 2:** Study sites and soil profiles on James Ross Island; a: St. Marta Cove (SMC). It is not covered with vegetation. A 90 cm deep soil profile was taken; b: soil profile St. Marta Cove (SMC). Scale of the tape measure is in cm; c: study site Brandy Bay (BB) is close to snowfield. It is not covered with vegetation. A 60cm soil profile was taken; d: Soil profile Brandy Bay (BB). Scale of the tape measure is in cm.

**Figure 3:** Images of micromorphological featuress found at Brandy Bay (BB) and St. Marta Cove (SMC). Pictures were taking using plane polarized light (ppl) and crossed polarizers (xpl). (a) BB III: highly separated lenticular platy microstructure, platy aggregates are indicated by green dotted lines, lenticular ms is indicated by black dotted lines, 2.5x, ppl; (b) SMC IV: moderately separated lenticular platy microstructure, indicated by black dotted lines, 2.5x, ppl; (c) SMC I: coarse monic microstructure, 2.5x, ppl; (d) BB II: chitonic c/f-related distribution and thin link cappings (li) on quartz grains, 20x, ppl; (e) BB III: weathered rock fragment covered by silty capping (cap) and also showing a thick pendent (pen) consisting of silty material and mineral grains, 10x, ppl; (f) SMC I: strongly weathered sandstone fragment with former boundaries, indicated by red dotted line, still visible by capping (cap), 5x, ppl; (g) SMC I: weathered volcanic rock fragment with distinct pellicular alteration pattern, 5x, ppl; (h) BB II: weathered and broken volcanic rock fragment with internal volcanic glass and covered by a thin clay capping (cap),(110-120μm), 2.5x, ppl; (i) SMC I: weathered volcanic rock fragment with feldspar phenocrysts; covered by a dusty clay-silt capping (80-100 μm) (cap), 2.5x, ppl; (i) SMC I:; usage of crossed polarizers makes it possible to tell external coating (cap) from altered internal material, border indicated by grey dotted line, 2.5x, xpl.

**Figure 4:** Relative abundances of phyla and bacterial 16S rRNA qPCR gene abundances of soil profiles from Brandy Bay (BB) and St. Marta Cove (SMC) on James Ross Island, Antarctica. Triplicates are merged. Only phyla with a relative abundance of at least 5% at a given site are shown. The remaining phyla are summarized as "Others".

**Figure 5:** Heatmap based on the relative abundances of the observed operational taxonomic units (OTUs) in soil profiles from Brandy Bay (BB) and St. Marta Cove (SMC) on James Ross Island, Antarctica. Only OTUs with a relative abundance of at least 3% in a given sample were included. Samples as well as OTUs were clustered using average linkage hierarchical clustering.

**Figure 6:** Canonical correlation analysis of the microbial composition of soil profiles from Brandy Bay (BB; black symbol) and St. Marta Cove (SMC; yello symbol) based on Bray-Curtis dissimilarities of the OTU data and its associated environmental parameters. If the Bonferroni corrected p-value was below 0.05, a given environmental parameter was included in the visualization. The amounts of chloride, sulfate, silt, Ca and TOC contents, and the $Fe_d/Fe_t$ ratio explained 49.9% of the microbial community composition.