# Peer review of "Pedogenic and microbial interrelation in initial soils under"

_Biogeosciences, 2018_

## Referee Comment (RC1) · Anonymous Referee #1 · 6 Jan 2019

In this work, Meier et al present a detailed investigation of two soil profiles from James Ross Island near the Antarctic Peninsula. They use standard techniques to analyse soil physicochemistry and microbial communities of the sites.

Overall, the manuscript is well-written and methodologically sound. The introduction provides an effective summary of what is known about how physicochemical conditions affect soil properties and microbial communities. The site description, methods, and results are clear and appropriate. The discussion brings the manuscript together, considering how the soil properties affect the microbial communities, and vice-versa.

[Figure]

The manuscript is extremely detailed.

I have two major suggestions for how the manuscript can be improved:

1. The authors should dedicate more discussion to the energy sources of the community

While the paper is generally very detailed, in my opinion more focus needs to be spent on the potential energy sources for the community. The cell counts observed are high for soils with such low organic carbon content.

Could inorganic energy sources such as atmospheric hydrogen, atmospheric CO, and ammonia potentially be sustaining this community? The authors mention that Actinobacteria were present, but other H2-scavenging phyla (Acidobacteria, Chloroflexi) and CO-scavenging phyla (Proteobacteria, Chloroflexi) are known.

It is also mentioned that potential ammonia-oxidising Thaumarchaeota are present in the community. Based on the physicochemical analysis, how much ammonia is available to sustain them?

It is also not clear, based on the results or figures, how abundant Cyanobacteria and algae were in the community. Can the authors dedicate a few sentences in the results to clarifying this? It is stated that phototrophs were 'nearly absent', but it would be more informative to state their relatively abundance (even if tiny). It is stated that chloroplast reads were removed, so presumably some chloroplasts were detected.

2. The authors should modify and consolidate the figures and possibly tables

The figures are not always as informative as the text. It is not entirely clear, based on the figure or legend, what the satellite image of Figure 1 and how this relates to the inlet. Could this figure be modified?

For Figure 2 to 5, could these photographs be amalgamated into a single multi-panel figure given they show similar things?

For Figure 8, while the heatmap is a useful summary, the odd colouring makes it hard to see trends. Could the authors modify this to increase the contrast and make more abundant OTUs darker than lighter. OTUs with 0% relative abundance should be white rather than navy blue.

In addition, some of the tables may be more suited for supplementary material.

I also have several minor suggestions:

L91-93: I disagree with this assessment. Most studied topsoils in Antarctic ice-free regions harbour diverse microbial communities with 16S rRNA gene counts exceeding 107. L82: Please change 'proofing' to 'proving' L99: Clarify what is meant by 'ornithogenic' soil given it is a specialised term L139-143: As this sentence is quite complicated, I recommend breaking it up into two: "These soils are not influenced by vascular plants, sulfides, and penguin rookeries. Our study aims to identify major soil and microbiological properties by combining pedochemical and micromorphological methods with microbial community studies based on high throughput sequence analyses." L500: Consider modifying 'laboratory' to 'study site'. L659: Please change 'fixate' to 'fix'

---

## Short Comment (SC1) · 14 Jan 2019

Dear referee,

Many thanks for the helpful and constructive comments on our paper. All comments and suggestions clearly contribute to an improvement of the manuscript, we are happy to implement them in their entirety. At the moment, we are adapting the figures according to your suggestions. We will then implement these changes together with the other referee and community comments as soon as we have received all comments

and remarks.

Sincerely, Lars Arne Meier

---

## Referee Comment (RC2) · Anonymous Referee #2 · 11 Feb 2019

The study aims at linking microbiological properties and their role in soil formation in the absence of plants. The study was performed at James Ross Island, Antarctica, where no vascular plants are occurring, and the authors identified two plots with different precipitation and sea spray input. Besides the performing microbial community studies based on high throughput sequence analyses, soils were investigated by pedochemical and micromorphological methods.

The manuscript fits well into the topic of Biogeosciences, and it presents new and interesting data on bacterial taxa of these soils, depending on soil environmental conditions

on the one hand and having a possible contribution to soil development on the other hand.

However, I have a problem that the intention of the manuscript is not clearly presented. From the introduction, one may understand that the manuscript is devoted to: - increase the general understanding of soils developed in the transitional zone of the eastern APR (l. 109-111), - add to the understanding of drivers of soil microbial diversity in high latitude soils (l. 125-126), - perfrom micromorphological studies on soils of the eastern APR (l. 132-134).

At the end of the introduction it appears that it is all a little bit (l. 139-143). Further, the mentioned goals are not embedded into a theoretical framework. This makes it a bit hard to prepare the potential reader of what can be learned by reading the manuscript, which goes beyond a list of microorganisms. Here, the authors may consider reworking the introduction incl. the objectives chapter.

A further problem that I encounter is that only two profiles are compared. I understand that at such regions of the world, it is often not possible to carry out a longer-term field study. But one must be aware that this is not a very solid basis for identifying cause-and-effect relations between the soil environment and the microbiota. Multivariate statistics could be performed, because the soil increments were considered as being independent form each other (if I understand the Bray-Curtis dissimilarity right). But at the other hand the authors also reported of water and solute flow through the profiles, thus linking the different horizons. But I think that this problem can be solved by a more careful discussion.

Abstract Also in the Abstract the goal of the study is written only in a quite vague manner. It is not clear, how the lee and luv position should impact the soil development? Was it the different input of salts with sea spray? Also the rest of the abstract is quite vague. E.g., what are the changes in soil microstructure below 20 cm depth and what is the potential impact on water availability and matter fluxes.

l. 53: Is it fair to say that the soils are dominated by bacterial taxa, when obviously no fungal taxa were investigated? But I believe as well that fungi most likely are of minor importance in these soils.

Introduction

Please, see my general comments given above.

The introduction largely emphasis the different soil forming conditions, primarily related to climate, at different regions of Antarctica. Even though there are usually no figures in the introduction, here I would suggest to show a map of Antarctica highlighting the different areas that are mentioned in the discussion (it can be a slightly modified version of the present Fig. 1). But, of course, this also depends on whether the editors will accept this suggestion.

l. 123-125: This sentence is not clear, actually sating that the microbial activity has an influence on the microbial composition . . . Please, rephrase.

Regional setting of James Ross Island, maritime Antarctica

Can be first subchapter of Material and Methods.

Material and Methods

l. 221: Please, indicate in what solution pH was measured.

l. 223-228: I do not understand how Cinorg (the abbreviation has not been introduced) can be measured by dry combustion after fumigation of the carbonates with HCl. I rather assume that Corg was measured and Cinorg was calculated by difference of Ctot and Corg. Otherwise, methods are properly described.

Results

l. 347: Why "virtually" unvegetated?

l. 357-360: Since this property was not identified in the field I would shift this paragraph

to the presentation of the micromorphological features.

l. 375-376: Present the TIC content as mg g-1. How can a TIC content transform to a TOC content? Consider rewording.

l. 378-380: Is there any explanation for the very low C/N ratios, most often much lower than in microbial biomass?

l. 395: Move this sentence to the beginning of the paragraph.

Discussion

In l. 192 a strong wind ablation was mentioned at BB. What is the role of the stronger ablation of fine material at BB on the chemical soil parameters? Can the selective erosion of a particular particle size blurr the results of the different weathering indices?

l. 499-501: I would rewrite the sentence "Due to the absence of vascular plants, the ice-free area of JRI is a pristine laboratory and offers the exceptional opportunity to improve our understanding of the interrelations between soil formation and microbiological properties" as "The JRI offers an exceptional opportunity to improve our understanding of the interrelations between soil formation and microbiological properties in the absence of plants".

l. 512-513: Present TOC and N contents as mg g-1.

l. 516-517: If low P contents refer to total P, then this cannot be taken to indicate a relative juvenility of the soils. Soils rather loose P with development than they gain. In the soils under study, there is no P input by birds and I assume that also the atmospheric P input is negligible.

l. 557-561: Here, I do not understand the line of argumentation.

l. 562-567: This part is quite speculative, but could have been easily proven. Why has Na not been leached before the total elemental analysis of the soil minerals? I cannot imagine the formation of stable secondary mineral phases entrapping Na.

l. 572-577: This is an important finding.

l. 585-609: Nice discussion based on micromorphology.

l. 610-674: The discussion on the different taxa is well written, and it is a good message that this initial stage of soil development, chemolithoautotrophic lifestyles plays an important role for the generation of biomass and initial accumulation of soil organic carbon and nitrogen (even though this finding is not really new). But might be this offers also a good opportunity for an introduction, in order to base it better on a conceptional background.

---

## Short Comment (SC2) · 21 Feb 2019

Dear Referee #2,

I also want to thank you for your thoughtful comments on our paper. All comments aim to improve our manuscript, make the statements really clear and present them precisely to the reader. We will change the addressed parts, as proposed and repeat and add some analyses to clarify the points raised. After the Open Discussion phase is over, we will implement the changes mentioned above.

Sincerely, Lars Arne Meier

---

## Author Comment (AC1) · 26 Apr 2019

**Response to reviewer comments on manuscript bg-2018-488: "From substrate**
**to soil in a pristine environment – pedochemical, micromorphological and**
**microbiological properties from soils on James Ross Island, Antarctica"**

We would like to thank the referees for their helpful and constructive comments, which
greatly helped to improve our manuscript. We have prepared a response where we
account for all points raised by the referees, as described below. We show the referees'
comments in grey text, while our responses are formatted as standard text. Line
indications refer to the changes in the revised manuscript.

**Anonymous Referee #1**

First, we would like to thank you for taking the time to review our manuscript. We
were glad about the positive and constructive feedback on our work.

1. The authors should dedicate more discussion to the energy sources of the
community
While the paper is generally very detailed, in my opinion more focus needs to be spent
on the potential energy sources for the community. The cell counts observed are high
for soils with such low organic carbon content.
Could inorganic energy sources such as atmospheric hydrogen, atmospheric CO, and
ammonia potentially be sustaining this community? The authors mention that
Actinobacteria were present, but other $H_2$-scavenging phyla (Acidobacteria,
Chloroflexi) and CO-scavenging phyla (Proteobacteria, Chloroflexi) are known.
Due to the lack of organic carbon as well as the low amount of potential phototrophic
organisms in the soils, we suspect inorganic energy sources to be crucial to sustain
the microbial ecosystem. This is supported by our data (Tab. S6) and observations of
a variety of microorganisms (e.g. Acidiferrobacteraceae, potential ammona-oxidizing
Thaumarchaeota), which potentially use such energy sources (e.g. L655ff, L666 –
669).

To underline this, we changed the text accordingly:
"Organisms with the ability to use oxygenic photosynthesis to fixate $CO_2$, such as
cyanobacteria, were nearly absent in the investigated soils. Low abundances of

Cyanobacteria are a common observation for Antarctic soil habitats (Ji et al., 2016). Due to the lack of phototrophic organisms and organic carbon, inorganic compounds and metabolic pathways utilizing those may have a more pronounced role in sustaining the microbial ecosystem at this initial stage of the soils." (L. 685-690)

In addition, we added a more detailed discussion regarding the usage of atmospheric compounds as energy sources:

"Further, a part of the community could use atmospheric compounds as energy source. Atmospheric $H_2$, CO, and $CO_2$ are scavenged and used as an energy source by microorganisms, especially organisms associated with the phyla Actinobacteria, Chloroflexi, Acidobacteria, Planctomycetes, Verrucomicrobia, and Proteobacteria (Greening et al., 2015; Ji et al., 2017). Operational taxonomic units related to the phylum Actinobacteria and the associated orders Acidimicrobiales and Solirubrobacterales were highly abundant in the investigated soils." (L. 705-711)

It is also mentioned that potential ammonia-oxidising Thaumarchaeota are present in the community. Based on the physicochemical analysis, how much ammonia is available to sustain them?

Potential ammonia-oxidising Thaumarchaeota have been present throughout the investigated profiles, and even have shown relative high abundances of up to 12.9%. However, ammonia, nitrite and nitrate could not be quantified by ion chromatography in any sample indicating negligible amounts, as written in L386f. It is well known that energy sources are scarce in ice-free areas of Antarctica (Souza et al., 2014; Cary et al., 2010), and might be metabolized quite quickly when they become available (e. g. from degradation of microbial necromass), which would explain the very low amounts as revealed by ion chromatography.

With this in mind, we modified the part discussing the presence of AOA in the discussion as follows:

"However, ion chromatography showed that amounts of ammonia as well as nitrite and nitrate were negligible in both soils. Ammonia originating from necromass and products in the course of nitrification could be metabolized directly by the present community, so no accumulation of the different intermediates containing nitrogen takes place." (L. 699-702)

It is also not clear, based on the results or figures, how abundant Cyanobacteria and
algae were in the community. Can the authors dedicate a few sentences in the results
to clarifying this? It is stated that phototrophs were 'nearly absent', but it would be more
informative to state their relatively abundance (even if tiny). It is stated that chloroplast
reads were removed, so presumably some chloroplasts were detected.

We agree with the reviewer, that both numbers are informative for the reader and
therefore added detailed information on the filtered reads which reveals the number of
OTUs associated with both Cyanobacteria and chloroplasts. We also included a table
with this information in the supplementary data (Tab. S4). However, we assume that
active phototrophic organisms only occur in the uppermost layers of the investigated
soils and reads in the deeper layers originate from translocated and phototrophically
inactive organisms.

The text was changed as follows:
"In total, 19,732,536 reads were obtained after merging the forward and reverse reads,
demultiplexing, filtering, and deletion of chimeric and singleton sequences.
Additionally, reads of chloroplast-associated OTUs (36,573), mitochondria-associated
OTUs (1,117) as well as rare OTUs (OTUs with a relative abundance of <0.1% in every
sample; 4,287,382) were filtered, resulting in 15,407,464 reads (Table S4)." (L. 468-
472)

In addition, we included observations on potential photosynthetic organisms in the
results:
"Regarding potential phototrophic organisms in the investigated soils, the amount of
chloroplast-related reads was relatively low (<0.2%) in each sample, except for SMC
>50 cm (0.03% - 1.30%) and BB 0 – 5 cm (0.87% - 1.01%). Cyanobacteria-related
OTUs were rare and only showed low relative abundances in SMC 5 – 10cm (0.06%),
SMC 10 – 20cm (; 0.62%), SMC >50cm (0.04%)." (L. 485-489)

 2. The authors should modify and consolidate the figures and possibly tables

The figures are not always as informative as the text. It is not entirely clear, based on the figure or legend, what the satellite image of Figure 1 and how this relates to the inlet. Could this figure be modified?

Thank you very much for this remark. As the satellite image seems not to be able to reflect the characteristics of the working area, we have decided to replace the satellite image with a map of the area.

We changed figure 1 as follows:

[Figure]

For Figure 2 to 5, could these photographs be amalgamated into a single multi-panel figure given they show similar things?

Many thanks. Combining the images into one multi-panel figure makes the chapter more compact and helps the reader to get the important information about the study sites.

We changed figure 2 as follows:

[Figure]

For Figure 8, while the heatmap is a useful summary, the odd colouring makes it hard
to see trends. Could the authors modify this to increase the contrast and make more
abundant OTUs darker than lighter. OTUs with 0% relative abundance should be white
rather than navy blue.

The heatmap was modified as suggested. Lighter colors represent lower relative
abundances, whereas darker colors represent higher relative abundances. We hope
that this change improves the overall clarity of the presented data.

We changed the heatmap as follows:

[Figure]

In addition, some of the tables may be more suited for supplementary material.

Thank you very much for the valuable assessment. Since the results in Tables 1, 2, and 4 are discussed directly in the paper, we consider these tables as basic information for our discussion. Table 3, on the other hand, is the basis for the results presented in

Table 2. For this reason, we agree with the proposal and move table 3 to the supplementary material. Here it becomes the new Table S1.

L91-93: I disagree with this assessment. Most studied topsoils in Antarctic ice-free regions harbour diverse microbial communities with 16S rRNA gene counts exceeding

107.

We agree that topsoils in Antarctic ice-free regions harbor diverse microbial communities, which are adapted to the present conditions, as we mentioned in e.g. L

92f or L115-121. Our statement in L90ff was targeted at groups such as higher plants, and vertebrates.

To emphasize this, we rephrased the paragraph as follows:

"Due to environmental stressors such as very low temperatures, low water availability, frequent freeze-thaw cycles and limited organic nutrient contents, soils from continental Antarctica show limiting conditions for higher organisms (Cary et al., 2010).

However, diverse microbial communities thrive in a variety of Antarctic habitats, such as permafrost soils (Cowan et al., 2014)." (L. 95-99)

L82: Please change 'proofing' to 'proving'

We changed it as follows:

"Therefore, soil scientific investigations became a relevant topic in Antarctic research, proving that there are actually soils in Antarctica (Jensen, 1916) and identifying soil forming processes (Ugolini, 1964)." (L. 87-89)

L99: Clarify what is meant by 'ornithogenic' soil given it is a specialised term

Ornithogenic soils are well known in Antarctica. The World Reference Base for Soil

Resources (WRB, 2014) defines ornithogenic material (from Greek ornithos, bird, and genesis, origin) as material with strong influence of bird excrement which often has a high content of gravel transported by birds.

The surface of these soils consists often of an indurated guano crust and scattered pebbles are common, since the penguins use them for their nests. The guano acts as additional source of nutrients particularly N and P.

We added the following information:

"Local conditions determine nutrient availability in Antarctic soils (Prietzel et al., 2019). Ca, Mg, K and P contents are generally high in igneous and volcanic rocks, whereas P and N contents are highest in ornithogenic soils. Ornithogenic soils are well known in Antarctica. The World Reference Base for Soil Resources (WRB, 2014) defines ornithogenic material (from Greek ornithos, bird, and genesis, origin) as material, which is characterized by penguin deposits mainly consisting of guano and often containing a high content of gravel transported by birds (cf. Ugolini, 1972)". (L. 103-109).

L139-143: As this sentence is quite complicated, I recommend breaking it up into two: "These soils are not influenced by vascular plants, sulfides, and penguin rookeries. Our study aims to identify major soil and microbiological properties by combining pedochemical and micromorphological methods with microbial community studies based on high throughput sequence analyses."

Thanks for providing this helpful comment. We changed this part as follows:

"This setting enables an investigation of interdependencies particularly between prokaryotic life and soil properties, since the selected soils are not influenced by vascular plants, sulfides, and penguin rookeries.

With this, the main goal of our study is to identify major soil and microbiological properties in an extreme environment by combining pedochemical and micromorphological methods with microbial community studies based on high throughput sequence analyses. Thus, we will gain a better general understanding of (i) the main soil forming processes and (ii) the drivers of soil microbial community structure in the eastern APR. This addresses also the question, if the variance of pedogenic and microbiological properties are larger between depth increments within one profile (e.g. with different distances to the permafrost table) or between different soil profiles, i.e. due to different local environmental conditions." (L. 157-167)

L500: Consider modifying 'laboratory' to 'study site'.

We agree with that comment. The paragraph reads now as follows:

"James Ross Island offers an exceptional opportunity to improve our understanding of the interrelations between soil formation and microbiological properties in the absence of plants." (L. 524-526)

L659: Please change 'fixate' to 'fix'

We are glad to comply with this remark. We changed this sentence as follows:

"Microorganisms can be seen as the primary pioneers of nutrient-poor environments such as Antarctic soils, and were shown to have the genetic potential to fix C and N

(Cowan et al., 2011; Niederberger et al., 2015), thus increasing C and N contents of these oligotrophic soils." (L. 693-696)

**Additional Literature:**

**Greening, C., Constant, P., Hards, K., Morales, S. E., Oakeshott, J. G., Russell, R.**
**J., Taylor, M. C., Berney, M., Conrad, R., and Cook, G. M.**: Atmospheric
Hydrogen Scavenging: From Enzymes to Ecosystems, Appl. Environ.
Microbiol., 81, 1190-1199, 2015.
**Ji, M., van Dorst, J., Bissett, A., Brown, M. V., Palmer, A. S., Snape, I., Siciliano,**
**S. D., and Ferrari, B. C.**: Microbial Diversity at Mitchell Peninsula, Eastern
Antarctica: A Potential Biodiversity "Hotspot", Polar Biology, 39, 237-249, 2016.
**Prietzel, J., Prater, I., Colocho Hurtarte, L. C., Hrbáček, F., Klysubun, W., and**
**Mueller, C. W.**: Site Conditions and Vegetation Determine Phosphorus and
Sulfur Speciation in Soils of Antarctica, Geochimica et Cosmochimica Acta, 246,
339-362, https://doi.org/10.1016/j.gca.2018.12.001, 2019.
**Ugolini, F. C.**: Ornithogenic Soils of Antarctica, in: Antarctic Terrestrial Biology, edited
by: Llano, G. A., 1972.

---

## Author Comment (AC2) · 26 Apr 2019

**Response to reviewer comments on manuscript bg-2018-488: "From substrate**
**to soil in a pristine environment – pedochemical, micromorphological and**
**microbiological properties from soils on James Ross Island, Antarctica"**

We would like to thank the referees for their helpful and constructive comments, which
greatly helped to improve our manuscript. We have prepared a response where we
account for all points raised by the referees, as described below. We show the referees'
comments in grey text, while our responses are formatted as standard text. Line
indications refer to the changes in the revised manuscript.

**Anonymous Referee #2:**

Before answering the individual comments, we would like to thank the referee for taking
a constructive and critical look at our manuscript.

However, I have a problem that the intention of the manuscript is not clearly presented.
From the introduction, one may understand that the manuscript is devoted to: -
increase the general understanding of soils developed in the transitional zone of the
eastern APR (l. 109-111), - add to the understanding of drivers of soil microbial
diversity in high latitude soils (l. 125-126), - perfrom micromorphological studies on
soils of the eastern APR (l. 132-134).

At the end of the introduction it appears that it is all a little bit (l. 139-143). Further, the
mentioned goals are not embedded into a theoretical framework. This makes it a bit
hard to prepare the potential reader of what can be learned by reading the manuscript,
which goes beyond a list of microorganisms. Here, the authors may consider reworking
the introduction incl. the objectives chapter.

Many thanks for your comment. We completely rewrote the introduction according to
you comment and changed almost the full introduction as follows:

"Therefore, soil scientific investigations became a relevant topic in Antarctic research,
proving that there are actually soils in Antarctica (Jensen, 1916) and identifying soil
forming processes (Ugolini, 1964)." (L. 87-89)

"However, diverse microbial communities thrive in a variety of Antarctic habitats, such
as permafrost soils (Cowan et al., 2014)." (L. 97-99)

"Local conditions determine nutrient availability in Antarctic soils (Prietzel et al., 2019).

Ca, Mg, K and P contents are generally high in igneous and volcanic rocks, whereas

P and N contents are highest in ornithogenic soils. Ornithogenic soils are well known in Antarctica. The World Reference Base for Soil Resources (WRB, 2014) defines ornithogenic material (from Greek ornithos, bird, and genesis, origin) as material, which is characterized by penguin deposits mainly consisting of guano and often containing a high content of gravel transported by birds (cf. Ugolini, 1972)." (L. 103-109)

"At the microscale, microbial activity such as photosynthesis and nitrogen fixation has a distinct influence on soil chemical parameters, e.g. the increase of carbon and nitrogen contents in oligotrophic soils (Ganzert et al., 2011; Cowan et al., 2011;

Niederberger et al., 2015). In return, these changes in soil characteristics affect microbial community composition." (L. 132-136)

"Since most of the non-lichenized Antarctic fungi are known to be decomposers and their abundance and distribution is limited by plant derived nutrients, and bio-available

Carbon (Arenz et al., 2011), the focus of this study lies on the prokaryotic interplay with soil characteristics and soil formation." (L 137-140)

"We selected two different soils, representing coastal soils and inland soils of James

Ross Island, developed on similar substrate and at similar topographic positions, but differing in local climate conditions and nutrient contents due to their relative position towards the mainly SW-winds. The western study site (Brandy Bay –BB) is located in a windward position and is highly influenced by sea spray, while the eastern study site (Santa Martha Cove – SMC), located behind a mountain range, is located in a leeward position (Prietzel et al., 2019). This setting enables an investigation of interdependencies particularly between prokaryotic life and soil properties, since the selected soils are not influenced by vascular plants, sulfides, and penguin rookeries.

With this, the main goal of our study is to identify major soil and microbiological properties in an extreme environment by combining pedochemical and micromorphological methods with microbial community studies based on high throughput sequence analyses. Thus, we will gain a better general understanding of (i) the main soil forming processes and (ii) the drivers of soil microbial diversity community structure in the eastern APR. This addresses also the question, if the variance of pedogenic and microbiological properties are larger between depth increments within one profile (e.g. with different distances to the permafrost table) or between different soil profiles, i.e. due to different local environmental conditions." (L.

151-167)

A further problem that I encounter is that only two profiles are compared. I understand that at such regions of the world, it is often not possible to carry out a longer-term field study. But one must be aware that this is not a very solid basis for identifying cause- and-effect relations between the soil environment and the microbiota. Multivariate statistics could be performed, because the soil increments were considered as being independent form each other (if I understand the Bray-Curtis dissimilarity right). But at the other hand the authors also reported of water and solute flow through the profiles, thus linking the different horizons. But I think that this problem can be solved by a more careful discussion.

Of course, we agree that the inclusion of additional soil profiles would increase the (statistical) power of our analysis. However, since this is not possible, at least for this paper, we followed your advice and rephrased the parts in the discussion based on our multivariate statistics and observations in a more careful fashion.

Following changes were made:

"In case of the pedogenic oxide ratios, 12.5% of the total compositional variation could be explained, which indicates a correlation between the microbial community structure and weathering at this very initial stage of soil formation." (L. 595-597)

"For example, the amount and size of microaggregates have been shown to be important regarding prokaryotic colonization, leading to genetically distinct communities as well as cell densities in different size classes of aggregates (Ranjard et al., 2000). Thus, in addition to chemophysical environmental parameters, which shape the overall prokaryotic community, the microstructure of the initial soils could have a substantial influence on species distribution." (L. 664-668)

We also added the following paragraph to better explain how we applied statistics:

"Multivariate statistics were performed for soil depth increments, which we considered to be independent. However, when processes are discussed that link soil horizons, e.g. water and solute flow through the profiles, we account for the limited number of two soil profiles with great care. We could not detect any environmental factors that increase or decrease the correlation between the chosen depth increments" (L. 628-

632)

Also in the Abstract the goal of the study is written only in a quite vague manner. It is not clear, how the lee and luv position should impact the soil development? Was it the different input of salts with sea spray? Also the rest of the abstract is quite vague. E.g., what are the changes in soil microstructure below 20 cm depth and what is the potential impact on water availability and matter fluxes.

Many thanks for this comment. We rewrote the abstract as follows:

"James Ross Island (JRI) offers the exceptional opportunity to study microbial driven pedogenesis without the influence of vascular plants or faunal activities (e.g. penguin rookeries). In this study, two soil profiles from JRI (one at St. Martha Cove - SMC, and another at Brandy Bay - BB) were investigated, in order to gain information about the initial state of soil formation and its interplay with prokaryotic activity, by combining pedological, geochemical and microbiological methods. The soil profiles are similar in respect to topographic position and parent material but are spatially separated by an orographic barrier and therefore represent windward and leeward locations towards the mainly south-westerly winds. These different positions result in differences in electric conductivity of the soils caused by additional input of bases by sea spray at the windward site, and opposing trends in the depth functions of soil pH and electric conductivity. Both soils are classified as Cryosols, dominated by bacterial taxa such as

Actinobacteria, Proteobacteria, Acidobacteria, Gemmatimonadates and Chloroflexi. A

shift in the dominant taxa was observed below 20 cm in both soils as well as an increased abundance of multiple operational taxonomic units (OTUs) related to potential chemolithoautotrophic Acidoferrobacteraceae. This shift is coupled with a change in microstructure. While single/pellicular grain microstructure (SMC) and platy microstructure (BB) is dominant above 20 cm, lenticular microstructure is dominant below 20 cm at both soils. The change in microstructure is caused by frequent freeze- thaw cycles and a relative high water content and goes along with a development of the pore spacing and is accompanied by a change in nutrient content. Multivariate statistics revealed the influence of soil parameters such as chloride, sulfate, calcium and organic carbon contents, grain size distribution, and pedogenic oxide ratios (POR)

on the overall microbial community structure and explained 49.9% of its variation. The correlation of the POR with the compositional distribution of microorganisms as well as the relative abundance certain microorganisms such as potentially chemolithotrophic Acidiferrobacteraceae-related OTUs could hint on an interplay between soil forming processes and microorganisms."(L. 42-67)

l. 53: Is it fair to say that the soils are dominated by bacterial taxa, when obviously no fungal taxa were investigated? But I believe as well that fungi most likely are of minor importance in these soils.

Most of non-lichenized Antarctic fungi are decomposers, and their abundance and distribution is limited by plant-derived nutrients and bio-available carbon (Arenz et al., 2011). Due to the absence of plants and lichens, and the overall low organic carbon contents, we assume that microbial communities are dominated by prokaryots and especially bacteria.

To clarify this, we changed the text as follows:

"In this study, two soil profiles from JRI (one at St. Martha Cove - SMC, and another at Brandy Bay - BB) were investigated, in order to gain information about the initial state of soil formation and its interplay with prokaryotic activity, by combining pedological, geochemical and microbiological methods. (L. 44-47)

"Since most of the non-lichenized Antarctic fungi are known to be decomposers and their abundance and distribution is limited by plant derived nutrients, and bio-available Carbon (Arenz et al., 2011), the focus of this studies lies on the prokaryotic interplay with soil characteristics and soil formation." (L. 137-140)

The introduction largely emphasis the different soil forming conditions, primarily related to climate, at different regions of Antarctica. Even though there are usually no figures in the introduction, here I would suggest to show a map of Antarctica highlighting the different areas that are mentioned in the discussion (it can be a slightly modified version of the present Fig. 1). But, of course, this also depends on whether the editors will accept this suggestion.

Many thanks for this remark. We replaced the satellite image of figure 1 with the following map. We suggest to mention figure 1 (L. 148) in the introduction and leave it in the methods section, because we describe there the study area more precisely.

[Figure]

This sentence is not clear, actually sating that the microbial activity has an
influence on the microbial composition . . . Please, rephrase.
We agree and rephrased this part as follows:
"At the microscale, microbial activity such as photosynthesis and nitrogen fixation has
a distinct influence on soil chemical parameters, e.g. the increase of carbon and
nitrogen contents in oligotrophic soils (Ganzert et al., 2011; Cowan et al., 2011;
Niederberger et al., 2015). In return, these changes in soil characteristics affect
microbial community composition." (L. 132-136)
Regional setting of James Ross Island, maritime Antarctica
Can be first subchapter of Material and Methods.
We moved the chapter "Regional setting of James Ross Island, maritime Antarctica"
now as a new subchapter into the "Material and Methods" section.
l. 221: Please, indicate in what solution pH was measured.
EC and pH were measured in deionized water. Probably the wording was misleading.
Therefore, we substituted the word "solution" by "water".
We changed the sentence as follows:
"Values of pH and electric conductivity were measured from bulk samples < 2mm in
deionized water with a sample to water ratio of 1:2.5." (L. 235-237).
l. 223-228: I do not understand how Cinorg (the abbreviation has not been introduced)
can be measured by dry combustion after fumigation of the carbonates with HCl. I
rather assume that Corg was measured and Cinorg was calculated by difference of
Ctot and Corg. Otherwise, methods are properly described.
Thank you for the important remark. We replaced "C$_{inorg}$" with the more common term
"TIC". We also changed this part of the material and methods chapter to clarify this
procedure:
"Carbon (C) and nitrogen (N) contents of the bulk soils were analyzed by dry
combustion (Elementar CNS Vario Max Cube). 300 to 500mg per sample were
analyzed in duplicate. In Order to distinguish between the total organic carbon (TOC)
content and the total inorganic carbon (TIC), TIC was removed by acid fumigation after
Ramnarine et al. (2011). 100 mg of the milled bulk soil samples were moistened with

20 to 40 µl of deionized water and put into a desiccator together with 100ml of 37%

HCl. Afterwards, the samples were dried at 40°C. Finally, the samples were measured again by dry combustion (EuroVector EuroEA3000 Elemental Analyser) to obtain the

TOC content. TIC content was calculated: TIC $= C_{tot} -$ TOC." (L. 238-245)

Many thanks, we deleted "virtually". The sentence was changed as follows:

"Both sites were unvegetated by cryptogamic or vascular plants." (L. 366-367)

223 l. 357-360: Since this property was not identified in the field I would shift this paragraph

224 to the presentation of the micromorphological features.

We moved this paragraph as suggested:

We changed the units in mg $g^{-1}$ for TOC and TIC. "Transform" is a wrong word; we rewrote this sentence:

"The TIC content was low in both soils ranging between 0.1 and 0.3 mg $g^{-1}$ in SMC and between 0.7 and 2.0 mg $g^{-1}$ in BB. The TOC content ranges from 0.8-0.9 mg $g^{-1}$ for

SMC and from 1.4 and 2.6 mg $g^{-1}$ for BB and increased there slightly with depth." (L.

391-393)

236 l. 378-380: Is there any explanation for the very low C/N ratios, most often much lower

237 than in microbial biomass?

Long periods of atmospheric deposition of salts in soil surfaces, the lack of leaching in arid areas and insignificant biological turnover may lead to comparably high nitrogen contents (Bockheim, 1997; Barrett et al., 2007). In combination with the generally low

C contents, these relative high N contents might lead to C/N ratios that depart from biological stoichiometry. Similar C/N ratios have been observed in other Antarctic soil habitats (e.g. Ganzert et al., 2011, Arenz et al., 2011, Barrett et al., 2007), which indicates this to be a common observation in such environments.

I. 395: Move this sentence to the beginning of the paragraph.

We moved the sentence as suggested.

Many thanks for your questions. We assume that the enrichment of pebbles at BB protect the finer material beneath them. However, a selective erosion of distinct grains sizes cannot be excluded, at least before the enrichment of coarser pebbles at the soil surface took place. The effect of selective erosion of fine particles is shown by the weathering indices, with lower CIA values in the top centimeters of both soil profiles. At BB, the influence of salts from sea spray is pronounced, with highest Na and Mg contents in the topsoils. We discussed this result as a rejuvenation effect of the weathering indices by salt input (L 534-536).

Further, we added the following sentence to the results section:
"The amount of coarse material > 2mm was larger at the profile BB. Deflation processes led to a residual enrichment of larger grains and pebbles at the soil surface of both profiles. The permafrost table was not reached in both soil profiles, but ground ice was visible in a depth of 85cm at SMC." (L. 362-366)

Thank you very much, we follow your suggestion and wrote:
"James Ross Island offers an exceptional opportunity to improve our understanding of the interrelations between soil formation and microbiological properties in the absence of plants." (L. 524-526)

l. 512-513: Present TOC and N contents as mg g-1.

We changed the units as follows:

"The examined soils on JRI were characterized by low TOC (0.9-2.6mg g$^{-1}$) and low total nitrogen contents (approx. 0.4mg g$^{-1}$), which is common for Antarctic soil environments (e.g. Cannone et al., 2008), and relative high pH values (7.4- 8.6)." (L.

536-538)

l. 516-517: If low P contents refer to total P, then this cannot be taken to indicate a relative juvenility of the soils. Soils rather loose P with development than they gain. In the soils under study, there is no P input by birds and I assume that also the atmospheric P input is negligible.

Many thanks for this remark. We omitted "and P". (L 541)

l. 557-561: Here, I do not understand the line of argumentation.

To clarifiy our line of argumentation, we rephrased the paragraph as follows:

"Interestingly, the relative abundances of these taxa changed according to the degree of weathering. This could indicate a possible interrelation between the occurrence of these potential weathering-related organisms and the degree of weathering of

Antarctic soils. (L. 583-585)

l. 562-567: This part is quite speculative, but could have been easily proven. Why has

Na not been leached before the total elemental analysis of the soil minerals? I cannot imagine the formation of stable secondary mineral phases entrapping Na.

Thanks for this comment. We conduct the XRF analyses generally with the total soil material. Leaching in advance of this analysis might leach also other elements than Na and change the results in an incalculable way. We added the results for Na from ion chromatography to Table 1. The results show that the amount of Na is significantly higher in BB, which is most likely because of Na input by sea spray. Regardless of its origin, Na is detected by XRF and therefore taken into account for the calculation of the CIA. For this reason, we cannot rule out the possibility that the CIA values for the

BB location may be underestimated.

We adjusted the following sentences:

"Ion Chromatography results show that the Na content is significantly higher at BB. This difference is most likely caused by the increased input of salts due to sea spray, which is known to carry high amounts of Na (Udisti et al., 2012). Since the calculation of the CIA takes Na into account (Nesbitt & Young, 1982), the CIA values would be significantly higher if the additional input of sea salts could be excluded." (L. 587-591)

l. 572-577: This is an important finding.

Many thanks.

l. 585-609: Nice discussion based on micromorphology.

Thank you very much as well.

l. 610-674: The discussion on the different taxa is well written, and it is a good message that this initial stage of soil development, chemolithoautotrophic lifestyles plays an important role for the generation of biomass and initial accumulation of soil organic carbon and nitrogen (even though this finding is not really new). But might be this offers also a good opportunity for an introduction, in order to base it better on a conceptional background.

Many thanks. At your advice, we added this to the introduction as follows:

"At the microscale, microbial activity such as photosynthesis and nitrogen fixation has a distinct influence on soil chemical parameters, e.g. the increase of carbon and nitrogen contents in oligotrophic soils (Ganzert et al., 2011; Cowan et al., 2011; Niederberger et al., 2015). In return, these changes in soil characteristics affect microbial community composition." (L. 132-136)

**Additional Literature**

**Arenz, B., and Blanchette, R.**: Distribution and Abundance of Soil Fungi in Antarctica at Sites on the Peninsula, Ross Sea Region and Mcmurdo Dry Valleys, Soil Biology and Biochemistry, 43, 308-315, 2011.

**Barrett, J. E., Virginia, R. A., Lyons, W. B., McKnight, D. M., Priscu, J. C., Doran, P. T., Fountain, A. G., Wall, D. H., and Moorhead, D.**: Biogeochemical Stoichiometry of Antarctic Dry Valley Ecosystems, Journal of Geophysical Research: Biogeosciences, 112, 2007.

**Prietzel, J., Prater, I., Colocho Hurtarte, L. C., Hrbáček, F., Klysubun, W., and Mueller, C. W.**: Site Conditions and Vegetation Determine Phosphorus and

Sulfur Speciation in Soils of Antarctica, Geochimica et Cosmochimica Acta, 246,
339-362, https://doi.org/10.1016/j.gca.2018.12.001, 2019.
**Ranjard L., Poly F., Combrisson J., Richaume A., Gour-bière F., Thioulouse J.,**
**Nazaret S.**: Heterogeneous celldensity and genetic structure of bacterial pools
associatedwith various soil microenvironments as determined by enu-meration
and DNA fingerprint approach (RISA), Microbiol.Ecol., 39, 263–272, 2000.